# eIF4E1b is a non-canonical eIF4E protecting maternal dormant mRNAs

Laura Lorenzo-Orts [1✉], Marcus Strobl[1], Benjamin Steinmetz[1,3], Friederike Leesch [1,2], Carina Pribitzer [1], Josef Roehsner[1,2], Michael Schutzbier[1], Gerhard Dürnberger[1] & Andrea Pauli [1✉]

## Abstract

**Maternal mRNAs are essential for protein synthesis during oogenesis and early embryogenesis. To adapt translation to specific needs during development, maternal mRNAs are translationally repressed by shortening the polyA tails. While mRNA deadenylation is associated with decapping and degradation in somatic cells, maternal mRNAs with short polyA tails are stable. Here we report that the germline-specific eIF4E paralog, eIF4E1b, is essential for zebrafish oogenesis. eIF4E1b localizes to P-bodies in zebrafish embryos and binds to mRNAs with reported short or no polyA tails, including histone mRNAs. Loss of eIF4E1b results in reduced histone mRNA levels in early gonads, consistent with a role in mRNA storage. Using mouse and human eIF4E1Bs (in vitro) and zebrafish eIF4E1b (in vivo), we show that unlike canonical eIF4Es, eIF4E1b does not interact with eIF4G to initiate translation. Instead, eIF4E1b interacts with the translational repressor eIF4E-NIF1, which is required for eIF4E1b localization to P-bodies. Our study is consistent with an important role of eIF4E1b in regulating mRNA dormancy and provides new insights into fundamental post-transcriptional regulatory principles governing early vertebrate development.**

**Keywords** eIF4E1b; P-bodies; Translational Regulation; Maternal mRNAs; Zebrafish
**Subject Categories** Development; RNA Biology; Translation & Protein Quality

## Introduction

Eggs contain a large number of ribosomes and mRNAs that enable protein synthesis in the embryo. However, to maintain a state of quiescence, several mechanisms are in place to repress translation in the egg. Maternal mRNAs are translationally repressed by shortening of the polyadenine (polyA) tails (Subtelny et al, 2014; Lee et al, 2023). mRNAs with short polyA tails are not efficiently recognized by the polyA-binding protein (PABP) in the egg and thus cannot serve as templates for polyA-dependent translation (Xiang and Bartel, 2021). In addition, recent work has shown that maternal ribosomes associate with several factors that block key functional sites of the ribosome and contribute to their dormancy (Leesch et al, 2023).

Since transcription stops at the onset of meiosis during oogenesis, eggs must store proteins and RNAs for later use in the early embryo. In zebrafish, for example, zygotic genome activation (ZGA) does not begin until 3 h post fertilization (hpf), whereas in mouse it begins at ~24 hpf (Jukam et al, 2017; Vastenhouw et al, 2019). In contrast to somatic cells, where mRNA deadenylation leads to decapping and degradation (Passmore and Coller, 2022), mRNAs with short polyA tails are stable in the oocyte and early embryo (Voeltz and Steitz, 1998; Bhat et al, 2023; Lee et al, 2023). While several RNA-binding proteins, including Zar1, Zar1l/Zar2, MSY2/Ybx2, and Igf2bp3, which bind to untranslated regions (UTRs) or coding sequences of mRNAs, have been implicated in stabilizing maternal mRNAs, it is unclear how these proteins protect maternal transcripts with short polyA tails from decapping at the 5'-end (Gillian-Daniel et al, 1998; Rong et al, 2019; Medvedev et al, 2008, 2011; Ren et al, 2020).

In somatic cells, the mRNA cap-binding factor eIF4E plays a key role in regulating translation and mRNA stability. eIF4E interacts with the scaffolding protein eIF4G in the cytoplasm, which contributes to the formation of the heterotrimeric complex eIF4F, consisting of eIF4E, eIF4G, and the RNA helicase eIF4A (Gingras et al, 1999). eIF4F is essential for canonical (cap- and polyA-dependent) translation and can be inhibited by eIF4EBPs, which compete with eIF4G for eIF4E binding (Marcotrigiano et al, 1999). eIF4E also interacts with the P-body component eIF4ENIF1/4E-T to inhibit mRNA decapping (Räsch et al, 2020). Decapping requires eIF4E to dissociate from the mRNA cap, which is otherwise inaccessible to the decapping enzyme Dcp2 (Vilela et al, 2000; Schwartz and Parker, 2000). eIF4E interactions depend on the presence of eIF4E-binding motifs (consisting of YXXXXLΦ; where X is any amino acid and Φ is a hydrophobic residue) in eIF4E-binding proteins, including eIF4EG, eIF4EBP, and eIF4ENIF1 (Mader et al, 1995; Dostie et al, 2000). Vertebrates have evolved specific eIF4E classes (i.e., eIF4E2/4EHP and eIF4E3) that can perform unique functions, such as translational inhibition upon

[1]Research Institute of Molecular Pathology (IMP), Vienna BioCenter (VBC), 1030 Vienna, Austria. [2]Vienna BioCenter PhD Program, Doctoral School of the University of Vienna and Medical University of Vienna, Vienna, Austria. [3]Present address: Department of Biology, Institute of Molecular Systems Biology, ETH Zürich, 8093 Zurich, Switzerland. ✉E-mail: laura.lorenzo@imp.ac.at; andrea.pauli@imp.ac.at

ribosome collision and translational initiation upon stress, respectively (Juszkiewicz et al, 2020; Weiss et al, 2021). In invertebrates, certain eIF4Es play important roles in germline development (Henderson et al, 2009; Huggins et al, 2020; Shao et al, 2023). Here we report that the germline-specific eIF4E paralog, eIF4E1b, is a noncanonical eIF4E that regulates maternal mRNA repression and storage.

## Results

Given the function of eIF4Es in regulating translation and mRNA stability, we hypothesized that eIF4Es may also contribute to maternal mRNA dormancy. To explore this possibility, we analyzed the expression of zebrafish *eIF4Es* during oogenesis and embryogenesis. Vertebrates have evolved three eIF4E classes with different affinities for the mRNA cap (Joshi et al, 2005). Class I eIF4Es contain two conserved tryptophans that interact with the mRNA cap. In class II (eIF4E2/4EHP) and class III (eIF4E3), one of these tryptophans is substituted by a different amino acid (Phe, Leu, or Tyr in the case of eIF4E2; Cys in the case of eIF4E3), which reduces the affinity for the mRNA cap (Zuberek et al, 2007; Osborne et al, 2013). Moreover, class II and class III eIF4Es contain other amino acid substitutions that affect their interaction with eIF4Gs (in the case of eIF4E2) and eIF4EBPs (in the case of eIF4E3) (Joshi et al, 2004). Zebrafish have seven eIF4Es, four of which belong to class I (eIF4Ea, eIF4Eb, eIF4E1c, and eIF4E1b), two to class II (eIF4E2 and eIF4E2rs1) and one to class III (eIF4E3) (Fig. 1A,B and Appendix Fig. S1). Zebrafish class I eIF4Es share between 65 and 84% sequence identity with human eIF4E (Fig. 1B). eIF4Ea and eIF4Eb are the most similar to mammalian eIF4E and are likely the result of a gene duplication event in fish (Fig. 1A,B and Appendix Fig. S1) (Taylor et al, 2001). While eIF4E1c is specific to fish (Appendix Fig. S1) (Rao et al, 2023), eIF4E1b proteins are conserved in most vertebrates (Fig. EV1A and Appendix Fig. S1). Expression data, namely polyA-selected RNA-seq (Pauli et al, 2012; Cabrera-Quio et al, 2021) and tandem mass tag mass spectrometry (TMT-MS), show that eIF4E1b and eIF4E1c are highly expressed during zebrafish oogenesis and early embryogenesis (Figs. 1C,D and EV1B), which is consistent with previous reports from zebrafish (Robalino et al, 2004; Rao et al, 2023) and mouse (Evsikov et al, 2006; Guo et al, 2023; Yang et al, 2023).

To understand the physiological relevance of eIF4E1b, we generated two CRISPR/Cas9-based *eif4e1b* knockout mutants in zebrafish containing different deletions in the third exon of the locus, resulting in frameshifts leading to premature stop codons (Fig. EV2A–C). Most homozygous *eif4e1b* mutants developed into fertile males, and only a small proportion developed into infertile fish that morphologically resembled females but had gonads with tumor-like growth and no oocytes (Figs. 1E–G and EV2D–G). Importantly, ubiquitous expression of GFP-tagged eIF4E1b under the control of the *actb2* (*actin, beta 2*) promoter in homozygous *eif4e1b* fish rescued the defect in female development and resulted in fertile males and females (Figs. 1E and EV2E,F). Since oocytes are necessary to maintain a female sex in zebrafish (Dranow et al, 2013), the male bias observed in *eif4e1b* adults could be due to sex reversal. To investigate this, we used the *ziwi:eGFP* reporter to identify juveniles (1–2 cm long fish still lacking secondary sexual characteristics) that had started to develop as females based on a high GFP expression in the gonads (Leu and Draper, 2010; Dranow

et al, 2016). While both homozygous *eif4e1b* mutant and wild-type siblings showed putative females with high GFP expression in their gonads as juveniles, only wild types developed into adult females (Fig. EV2H,I), suggesting that loss of eIF4E1b causes sex reversal in zebrafish. In line with this, high GFP-expressing gonads from homozygous *eif4e1b* juveniles were either ovaries (with similar morphology as those isolated from wild types) or gonads containing differentiating sperm, the latter most likely representing ovaries in the process of transforming into testes (Fig. EV2J). We therefore conclude that eIF4E1b is required for female germline development in zebrafish.

Although eIF4E1b proteins belong to class I eIF4Es, they have been reported not to bind (Robalino et al, 2004) or to bind weakly (Minshall et al, 2007; Kubacka et al, 2015) to the mRNA cap, despite containing the two conserved tryptophan residues that are responsible for binding to the mRNA cap in canonical eIF4Es (Figs. 1A and 2A). To test the ability of eIF4E1bs to bind to the mRNA cap, we performed in vitro immunoprecipitation assays with $m^7G$-coated beads and bacterial lysates containing soluble His and MBP-tagged eIF4Es. As recombinant eIF4E1b from zebrafish was unstable in solution (Appendix Fig. S2A), we used mouse and human eIF4E1Bs for our in vitro studies (Fig. EV1A). We observed that human eIF4E1B binds to $m^7G$ with an affinity similar to that of eIF4Ea (Fig. 2B and Appendix Fig. S2B), which is 83% identical to human eIF4E and contains all the residues involved in mRNA cap binding (Fig. 1A,B). In support of the specificity of this interaction, the affinity of eIF4E1B for $m^7G$ was reduced when the tryptophans involved in mRNA cap binding in canonical eIF4E were mutated, suggesting that eIF4E1B binds to the mRNA cap in the same manner as eIF4E (Fig. 2A,B and Appendix Fig. S2B). To investigate whether eIF4E1B can also bind to the eIF4E-binding motifs of eIF4G, eIF4EBP1 and eIF4ENIF1 like other class I eIF4Es, we performed in vitro pulldown experiments with bacterial lysates (Fig. 2C,D and Appendix Fig. S2C–E). eIF4Ea and eIF4E1c bound to all tested eIF4E-binding motifs with similar affinities as murine eIF4E, although eIF4Ea showed a slightly higher affinity for eIF4ENIF1 (Fig. 2D). In agreement with previous studies using the mouse or human eIF4Es, we observed that zebrafish eIF4E3, but not eIF4E2, bind to eIF4G (Fig. 2D) (Joshi et al, 2004; Osborne et al, 2013; Weiss et al, 2021). Unlike other class I eIF4Es, neither mouse nor human eIF4E1B interacted with eIF4G, although they bound to eIF4EBP1 and eIF4ENIF1 (Fig. 2D and Appendix Fig. S2F). Taken together, our in vitro data show that, contrary to previous reports (Robalino et al, 2004; Minshall et al, 2007; Kubacka et al, 2015), eIF4E1Bs efficiently bind to the mRNA cap and to the eIF4E-binding motifs of eIF4EBP1 and eIF4ENIF1, yet do not interact with eIF4G and thus may not be promoting translation initiation like other class I eIF4Es.

The interaction of eIF4E with eIF4ENIF1 has been reported to trigger the localization of eIF4E to P-bodies and to repress translation (Ferraiuolo et al, 2005). Since eIF4E1B showed the highest affinity for eIF4ENIF1 among all eIF4Es tested, we investigated its subcellular localization. To this end, we generated transgenic zebrafish lines expressing GFP-tagged versions of eIF4Ea, eIF4E1c, and eIF4E1b under the control of the ubiquitously expressed *β-actin* promoter. GFP-tagged eIF4Es localized to the cytoplasm of zebrafish oocytes and were excluded from the Balbiani body (Figs. 2E,F and EV3A), a membraneless organelle composed of mitochondria, endoplasmic reticulum, and RNA that is

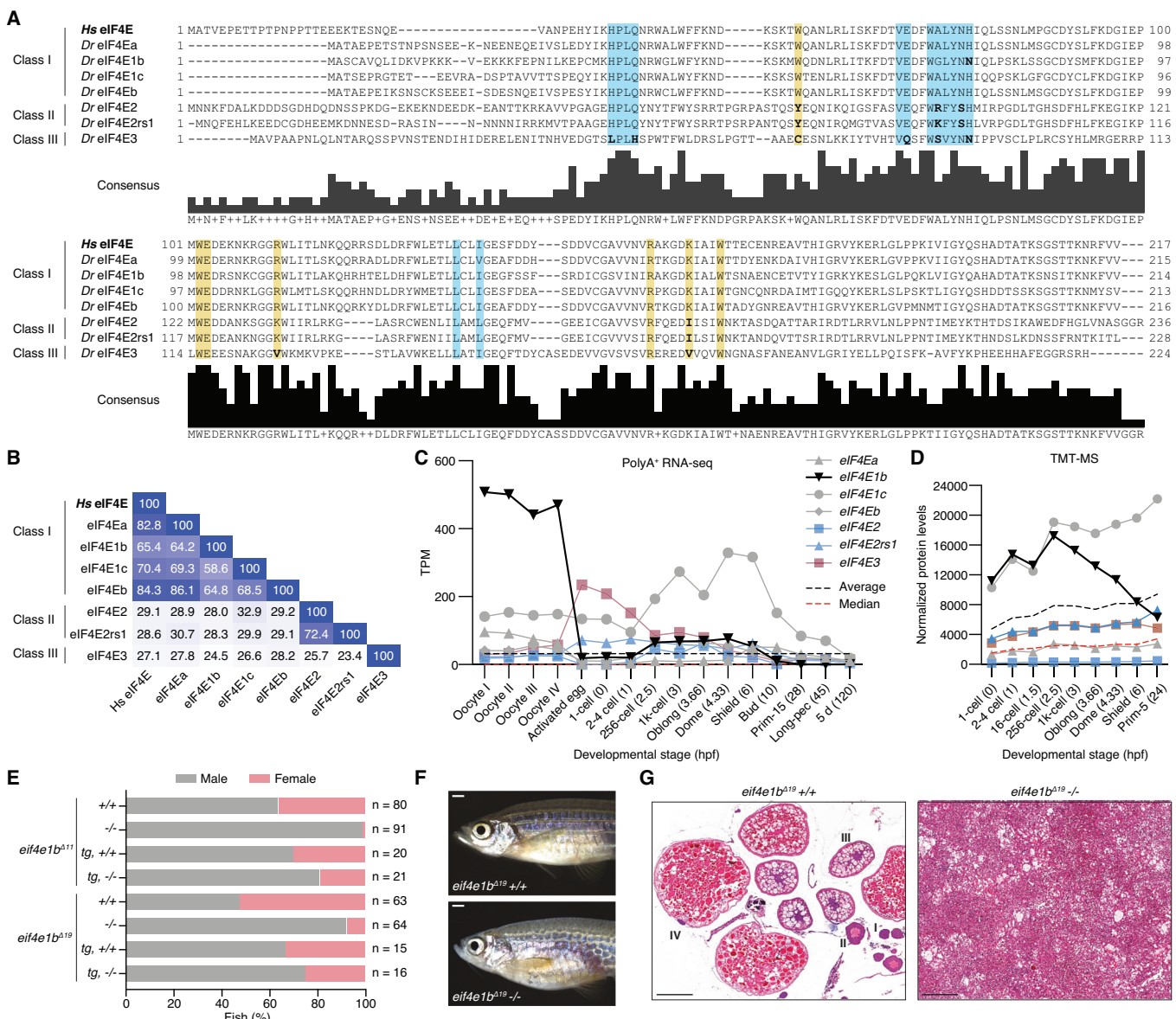

**Figure 1. eIF4E1b is a class I eIF4E protein with an essential role in zebrafish oogenesis.**

(A) Alignment of human (*Homo sapiens, Hs*) and zebrafish (*Danio rerio, Dr*) eIF4E proteins. Highlighted regions indicate the amino acids involved in the interaction with the mRNA cap (yellow) and eIF4E-binding motifs (blue); residues in the highlighted regions with a different polarity than human eIF4E are indicated in bold. (B) Percentage identity matrix of the proteins shown in (A). (C) The abundance of *eIF4E* transcripts during zebrafish oogenesis and embryogenesis based on published polyA-selected RNA-seq data (Pauli et al, 2012; Cabrera-Quio et al, 2021). TPM transcripts per million. *X* axis indicates developmental stages (hours post fertilization, hpf, in brackets). (D) eIF4E protein levels (represented with the same colors and symbols as in (C) during early zebrafish embryogenesis obtained by tandem mass tag mass spectrometry (TMT-MS), normalized to spike-in proteins. *X* axis indicates developmental stages (hpf in brackets). (E) Percentage of males and females of homozygous (−/−) and wild-type (+/+) siblings obtained from heterozygous *eif4e1b* incrosses (n = number of fish), as determined by secondary sexual characteristics. Expression of *3xflag-sfGFP-eIF4E1b* (*tg*) partially rescues the male bias observed in the two homozygous fish mutants. (F) Representative images of wild-type (top) and homozygous *eif4e1b* (bottom) female siblings (scale bar = 1 mm). (G) Hematoxylin and eosin staining of sectioned ovaries isolated from wild-type and homozygous *eif4e1b* fish. Mutant ovaries have no oocytes. Stages of oocyte development are indicated with roman numbers in the wild-type ovary sections. Scale bars = 200 μm. Source data are available online for this figure.

important for germline determination in zebrafish (Jamieson-Lucy and Mullins, 2019a). While eIF4Ea and eIF4E1c showed a diffuse cytosolic signal at all stages of oogenesis (Figs. 2E and EV3A), eIF4E1b formed cytoplasmic puncta in stage III oocytes (Fig. 2F), similar to the localization observed for the P-body component DDX6 in mouse oocytes (Flemr et al, 2010). In activated eggs and embryos, eIF4E1b localized to the cytosol and to cytoplasmic

granules colocalizing with P-body markers such as Dcp2, Ddx6, and Ybx1 (Figs. 2F and EV3B–D). These data suggest that eIF4E1b localizes to P-bodies during embryogenesis, in line with the interaction observed between eIF4E1b and eIF4ENIF1 in vitro.

Although eIF4E1b shares ~60% amino acid identity with other class I eIF4Es, our in vitro and in vivo data suggest that eIF4E1b does not function as a canonical eIF4E. We combined sequence

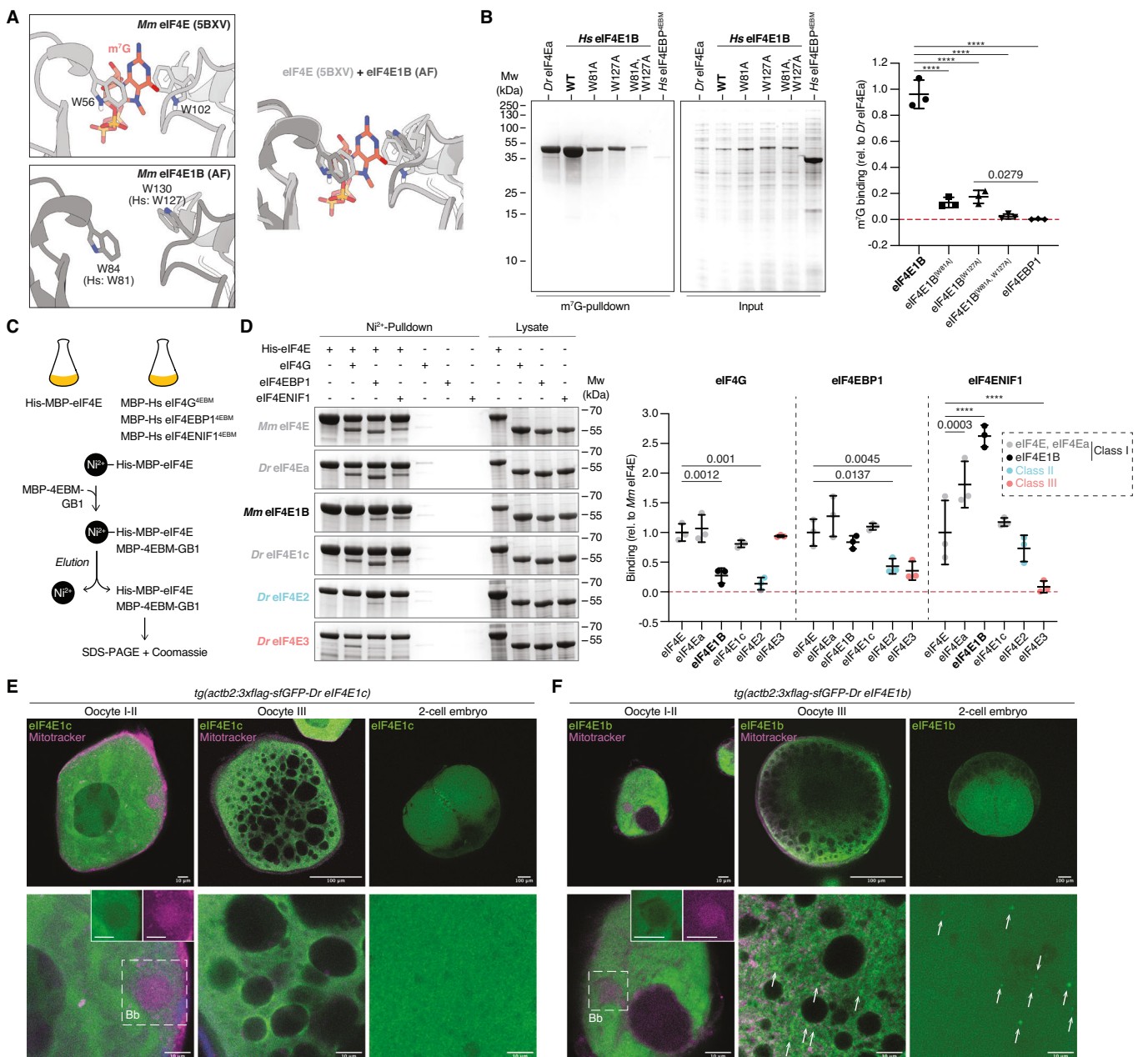

**Figure 2. eIF4E1b interacts with the mRNA cap, eIF4EBP1, and eIF4ENIF1 and localizes to cytoplasmic foci in oocytes and embryos.**

(**A**) eIF4E interaction with the mRNA cap (top, PDB-5BXV, Sekiyama et al, 2015) is mediated by two tryptophans that are conserved in eIF4E1B (bottom, AlphaFold (AF) prediction of mouse eIF4E1B). The superimposition of both structures is shown on the right, with eIF4E in light gray, eIF4E1B in dark gray, and m7G in red. (**B**) (Left) Coomassie-stained gels from immunoprecipitation assays with *E. coli* lysates containing zebrafish eIF4Ea and human eIF4E1B (wild-type and tryptophan mutants, see (**A**)) using m7G-coated beads. The eIF4E-binding motif (4EBM) of human eIF4EBP1 is used as a negative control. Quantification of eIF4E binding to m7G (relative to eIF4Ea) is shown on the right. (**C**) Scheme of in vitro pulldown assays. Lysates of *E. coli* cells expressing His- and MBP-tagged eIF4Es were incubated with Ni²⁺ beads. Lysates containing MBP-tagged 4EBMs of human eIF4G, eIF4EBP1, or eIF4ENIF1 (see Appendix Fig. S2C–E) were added to the beads. After elution, binding of 4EBMs to eIF4E was assessed by SDS-PAGE and Coomassie staining. (**D**) (Left) Coomassie-stained gels of pulldowns with mouse eIF4E, eIF4Es from zebrafish (eIF4Ea, eIF4E2, and eIF4E3), and mouse eIF4E1B with 4EBMs from eIF4G, eIF4EBP1, and eIF4ENIF1. Quantifications are shown on the right. (**E**, **F**) Confocal microscopy images of fixed transgenic zebrafish oocytes and embryos expressing 3xflag-sfGFP-eIF4E1c (**E**) and 3xflag-sfGFP-eIF4E1b (**F**). Mitochondria were stained with Mitotracker (in magenta). Images at two different magnifications are shown at the top and bottom (scale bars correspond to 100 and 10 μm, respectively). The Balbiany body (Bb) is indicated by a dashed box; individual channels are shown in boxes for the Bb. Cytoplasmic foci are highlighted with arrows. Data information: (**B**, **D**) $n = 3$ independent experiments. Significance was determined using two-way ANOVA (**B**) or one-way ANOVA (**D**) followed by Tukey's (**B**) or Dunnett's (**D**) multiple comparisons test (****$P$ value < 0.0001). Lines indicate mean with SD. (**B**, **D**) Predicted molecular weights (in kDa) are: *Dr* eIF4Ea, *Hs* eIF4E1B, *Mm* eIF4E1B, *Dr* eIF4E1c and *Dr* eIF4E2 = 65; *Mm* eIF4E = 66; *Dr* eIF4E3 = 64; *Hs* eIF4G[4EBM] and *Hs* eIF4ENIF1[4EBM] = 53; *Hs* eIF4EBP1[4EBM] = 52. Source data are available online for this figure.

alignments with available structural data and AlphaFold (AF) predictions (Mirdita et al, 2022; Evans et al, 2022) of mouse eIF4E paralogs to investigate the contribution of specific amino acids in determining eIF4E1B interactions. AF predicted an eIF4E1B structure very similar to eIF4E (Appendix Fig. S3A). The eIF4E-binding motifs of eIF4G, eIF4EBP, and eIF4ENIF1 have been reported to bind to the so-called dorsal and lateral surfaces of eIF4E (Fig. 3A) (Igreja et al, 2014; Sekiyama et al, 2015; Peter et al, 2015; Grüner et al, 2018). Mutation of the conserved Trp101 located on the dorsal surface of mouse eIF4E1B (Fig. 3B) abolished the interaction with eIF4EBP1 and eIF4ENIF1 (Fig. 3D and Appendix Fig. S3B), indicating a similar binding mode of eIF4E-binding motifs to the dorsal surfaces of eIF4E and eIF4E1B. Mutation of residues located on the lateral surface of mouse eIF4E1B (Fig. 3B) abolished eIF4EBP1 binding, whereas eIF4ENIF1 binding was only reduced (Fig. 3D and Appendix Fig. S3B), suggesting that additional residues in eIF4E1B stabilize the interaction with eIF4ENIF1. To investigate this further, we generated eIF4E1B-eIF4E chimeras (Fig. EV1A). Exchanging the N-terminal half of eIF4E1B (eIF4E-eIF4E1B) restored its ability to interact with eIF4G and decreased its affinity for eIF4ENIF1 (Fig. 3D and Appendix Fig. S3B), suggesting that the N-terminal half of eIF4E1B is important for defining eIF4E1B interactions. Based on AF predictions, we hypothesized that Lys108 of mouse eIF4E1B binds to eIF4ENIF1 (Fig. 3E), reminiscent of the interaction observed between Drosophila eIF4E and the fly ortholog of eIF4ENIF1, 4E-T (Appendix Fig. S3C) (Peter et al, 2015). In eIF4Es, an Asn is predicted to form an intramolecular amide bond with the residue equivalent to Lys108 in mouse eIF4E, Gln80, thereby preventing the interaction of this residue with eIF4ENIF1 (Fig. 3E). In line with this hypothesis, mutation of Lys108 to an opposite charge (Glu) or of Lys112 to an Asn (as in eIF4Es) reduced the affinity of eIF4E1B for eIF4ENIF1 (Fig. 3D and Appendix Fig. S3B). In addition to Lys112, we identified three other residues within the N-terminal half of eIF4E1Bs that are conserved in eIF4Es but differ in eIF4E1Bs (Figs. 3C and EV1A). However, mutation of all four residues to their equivalent amino acids in eIF4Es did not increase eIF4G binding, whereas it decreased the affinity for eIF4ENIF1 to a similar extent as mutation of Lys112 alone (Fig. 3D and Appendix Fig. S3B). Taken together, our in vitro data show that eIF4E1b proteins contain specific residues that interact with eIF4ENIF1.

eIF4ENIF1 has been reported to target eIF4E to P-bodies in human cells (Ferraiuolo et al, 2005). Since eIF4E1B interacts with eIF4ENIF1 in vitro and localizes to P-bodies in zebrafish embryos (in contrast to the cytosolic and nuclear localization of other class I eIF4Es with lower affinities for eIF4ENIF1; Figs. 2D,E and EV3B), eIF4ENIF1 may also determine the subcellular localization of eIF4E1b in zebrafish. Moreover, binding to the mRNA cap or to other proteins may also influence the assembly of eIF4E1b into P-bodies. To test these ideas, we expressed GFP-tagged wild-type or mutant versions of zebrafish eIF4Es via mRNA injections into 1-cell embryos and imaged their localization at 3 hpf (Fig. 3F,G). Localization of zebrafish eIF4E1b to P-bodies was not affected by mutation of Trp53 and Trp99, which are required for mRNA cap binding in vitro (Fig. 2A,B), or by deletion of its unstructured N-terminus, which is the region most distinct from eIF4Es (Figs. 1A and 3H and Appendix Fig. S3D). In contrast, mutation of residues located on the dorsal or lateral surface of eIF4E1b that are important for the interaction with eIF4EBP and eIF4ENIF1

abolished its accumulation in granules (Fig. 3H and Appendix Fig. S3D), indicating that eIF4E1b localization to P-bodies does not require binding to the mRNA cap but to other proteins. Notably, mutation of residues identified in our in vitro experiments as important for eIF4ENIF1 but not for eIF4EBP1 binding (see Fig. 3D) was sufficient to reduce the number of eIF4E1b foci in the embryo (Fig. 3H and Appendix Fig. S3D). Taken together, these data suggest that the ability of eIF4E1b to localize to P-bodies is dependent on eIF4ENIF1.

To gain insights into the eIF4E1b protein interactome, we performed immunoprecipitation (IP) followed by mass spectrometry experiments using anti-GFP beads and lysates from either wild-type or transgenic zebrafish oocytes and embryos expressing GFP-tagged eIF4E1b. Transgenic zebrafish expressing GFP-tagged eIF4E1c were used as a control because eIF4E1c is abundant during early embryogenesis (Fig. 1D) and has a similar affinity for eIF4G, eIF4EBP1, and eIF4ENIF1 as mammalian eIF4E in vitro (Fig. 2D). In oocytes and 8-cell embryos, eIF4E1b interacted with eIF4EBPs and P-body components, including eIF4ENIF1, Zar1l/Zar2, Ybx1, Lsm14/Rap55 proteins, and Ddx6, but not with eIF4Gs (Figs. 4A,B and EV4A and Datasets EV1 and EV2). Notably, all P-body components except eIF4ENIF1 interacted with eIF4E1b only in the presence of RNA (Fig. EV4C; Dataset EV2). At 3 hpf, during ZGA, eIF4E1b no longer associated with Zar1l, whose mRNA and protein levels decline after 1 hpf (Figs. 4C and EV4D,E; Dataset EV3). In line with our in vitro experiments, eIF4E1c interacted with eIF4Gs, eIF4EBPs, and eIF4ENIF1 in vivo (Figs. 4D–F and EV4B; Datasets EV1 and EV2). Moreover, translational factors belonging to the eIF3 complex were specifically enriched in the eIF4E1c IP at 3 hpf (Fig. 4F; Dataset EV3). These results suggest that eIF4E1c functions as a canonical eIF4E in vivo, promoting translation initiation, whereas eIF4E1b plays a role in P-bodies. While mRNA decapping components (e.g., Dcp1, Dcp2, and Edc4) localize to P-bodies (Greber et al, 2016) (Fig. EV3D), it is noteworthy that none of these proteins were detected in eIF4E1b IPs, despite their expression in zebrafish oocytes and embryos (Fig. EV4F–H). Consistent with the similar binding mode of eIF4E and eIF4E1B to the mRNA cap (Fig. 2A,B), our proteomics data suggest that eIF4E1b interferes with the binding of the decapping machinery to the mRNA cap (Fig. EV4I), as reported for canonical eIF4E in human cells and yeast (Schwartz and Parker, 2000; Räsch et al, 2020).

To identify which mRNAs are bound by eIF4E1b, we performed RNA immunoprecipitation (RIP) followed by sequencing in transgenic zebrafish 8-cell embryos (1.25 hpf) expressing GFP-tagged eIF4E1b or eIF4E1c. Differential expression analysis revealed 1206 and 1414 genes as specifically enriched in eIF4E1b and eIF4E1c RIPs, respectively ($P$ value < 0.005) (Figs. 5A and EV5A,B; Dataset EV4). Overexpression of eIF4E1b and eIF4E1c did not result in major changes in the transcriptome at 1.25 hpf, with 1.6% (363 transcripts) and 2.2% (500 transcripts) being differentially enriched in lysates from 8-cell embryos overexpressing eIF4E1b and eIF4E1c, respectively (Fig. EV5C,D; Dataset EV4). Approximately 34% and 20% of the transcripts upregulated in eIF4E1b or eIF4E1c overexpressing embryos were also enriched in the eIF4E1b or eIF4E1c RIP, respectively (Fig EV5E), suggesting a role for eIF4Es in stabilizing the mRNAs to which they bind. Genes enriched in the eIF4E1b RIP had GO-terms related to chromatin regulation, whereas genes enriched in the eIF4E1c RIP belonged to GO-terms associated with mRNA processing and export (Fig. 5B).

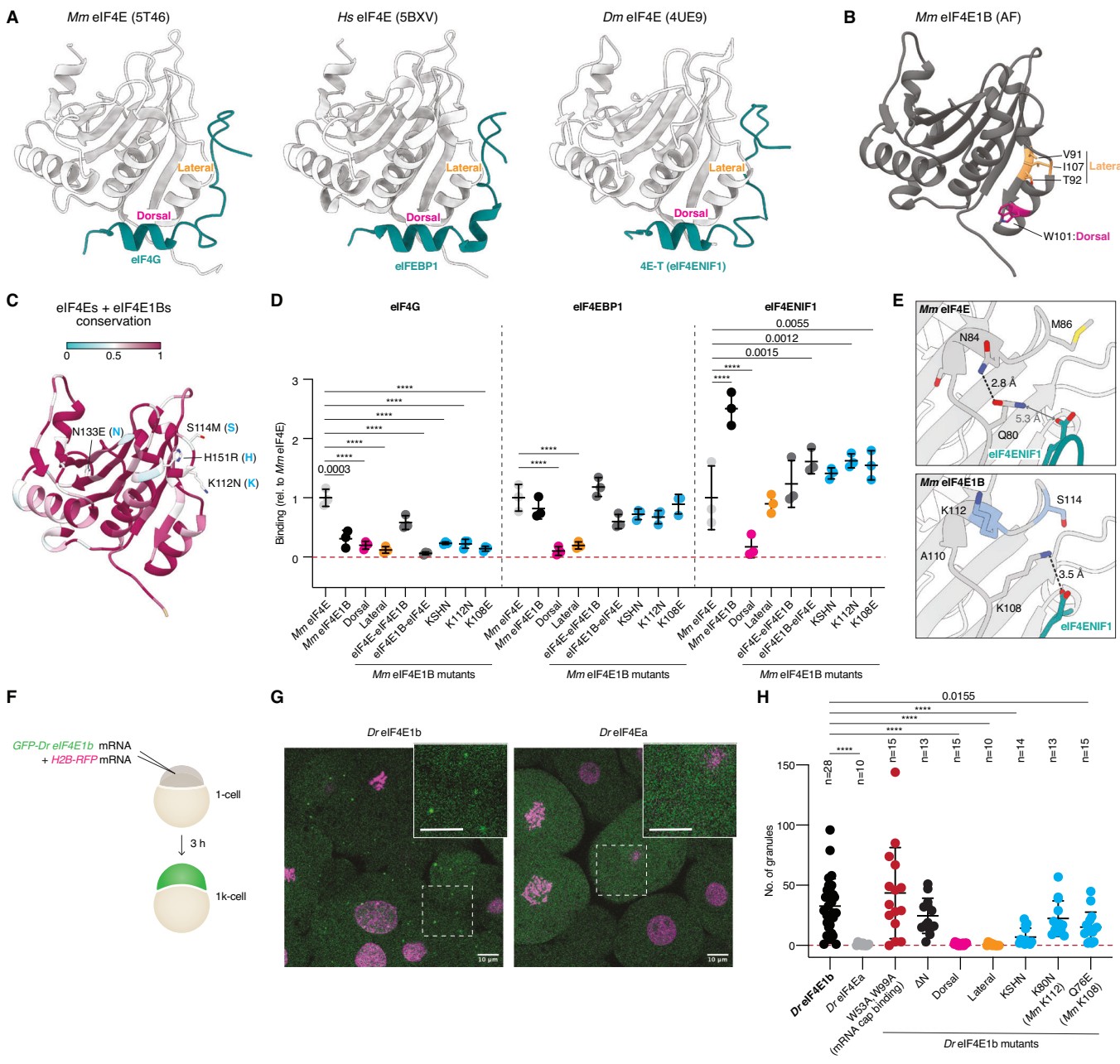

**Figure 3. Specific eIF4E1b residues mediate eIF4ENIF1 binding and localization to P-bodies in the embryo.**

(A) Structures of eIF4E proteins (in gray) bound to the eIF4E-binding motifs (in teal) of eIF4G (PDB-5T46; Grüner et al, 2016), eIF4EBP1 (PDB-5BXV; Sekiyama et al, 2015), and 4E-T/eIF4ENIF1 (PDB-4UE9; Peter et al, 2015). (B) AlphaFold (AF)-predicted structure of mouse eIF4E1B. Residues located at the dorsal and lateral surfaces are highlighted in magenta and orange, respectively. (C) The AF structure of mouse eIF4E1B is colored based on amino acid conservation among vertebrate eIF4E and eIF4E1B proteins (sequences in Fig. EV1A). Residues in the N-terminal half that differ in eIF4E1Bs but are conserved in eIF4Es are indicated (first residue: eIF4E1b; second residue: eIF4E). (D) Quantification of eIF4G, eIF4EBP1 and eIF4ENIF1 binding to mouse eIF4E1B wild-type and mutant proteins in pulldowns with *E. coli* lysates (see Appendix Fig. S3B; $n = 3$ independent experiments), compared to mouse eIF4E. eIF4E and eIF4E1B data are also plotted in Fig. 2D. KSHN refers to the residues highlighted in (C). (E) AF-predicted structures of mouse eIF4E (top) or eIF4E1B (bottom) in complex with the eIF4E-binding motif of human eIF4ENIF1. Distances are indicated; interactions are depicted with dashed lines. (F) Assay to test the contribution of specific amino acids in determining the subcellular localization of eIF4E1b in zebrafish embryos. mRNAs were co-injected into 1-cell embryos; embryos were imaged after 3 h. (G) Representative confocal microscopy pictures of live embryos transiently expressing GFP-tagged eIF4E1b or eIF4Ea (green) and H2B-RFP (magenta). Regions delimited by dashed boxes are shown at a higher magnification (scale bars = 10 μm). (H) The number of eIF4E-positive granules counted in three images taken at different positions of the embryo ($n$ = embryos). Data information: (D, H) significance was determined with two-way (D) or one-way (H) ANOVA followed by Dunnett's test (****$P$ value < 0.0001). Lines indicate mean with SD. Source data are available online for this figure.

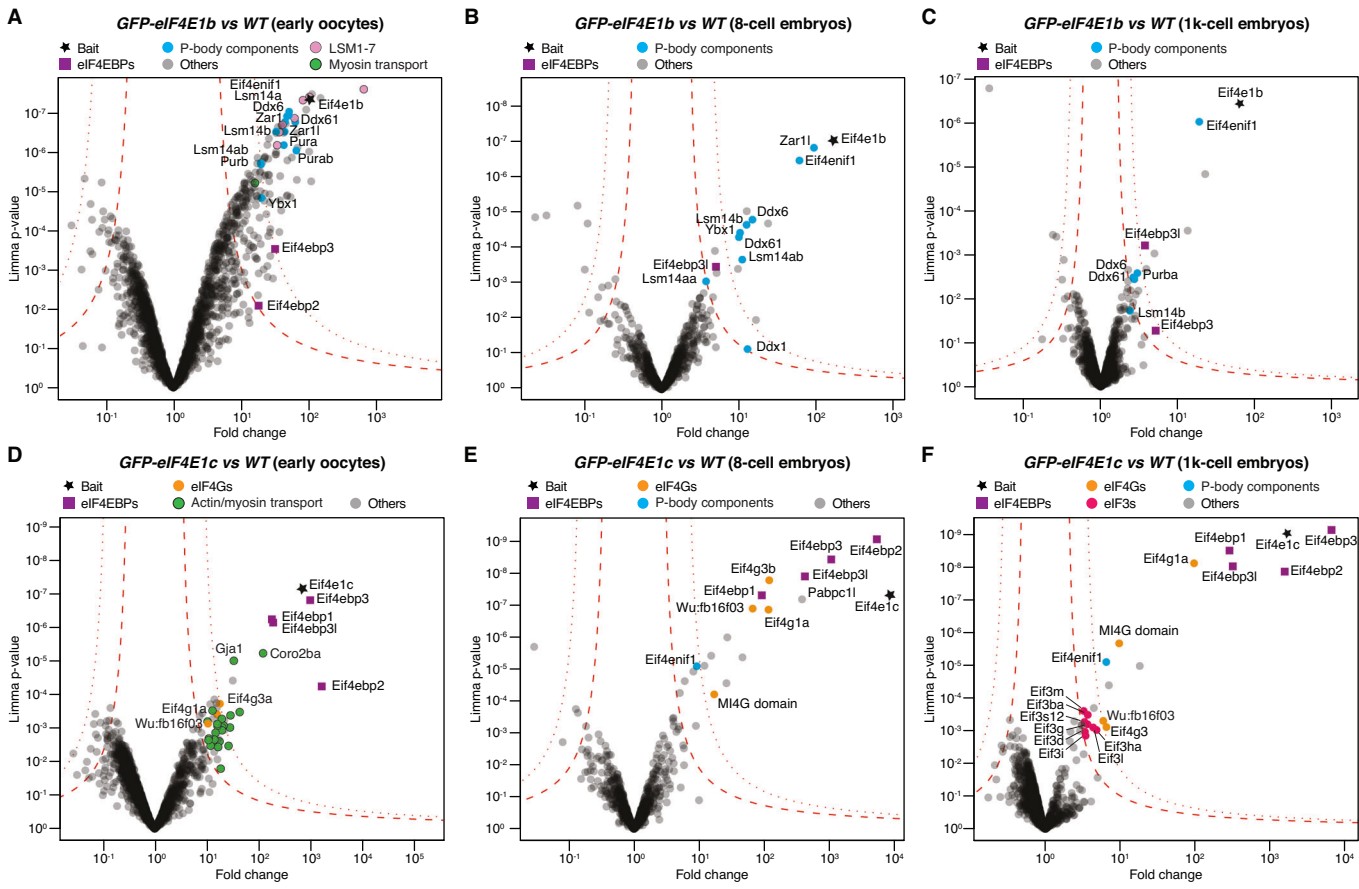

**Figure 4. eIF4E1b interacts with P-body components in zebrafish oocytes and embryos.**

Volcano plots of proteins enriched by immunoprecipitation followed by mass spectrometry (IP-MS) using GFP-tagged eIF4E1b (A–C) or eIF4E1c (D–F) as bait. Wild-type (WT) lysates were used to control for unspecific binding to the beads. Early oocytes correspond to oogonia and stage I–II oocytes. Permutation-based false discovery rates (FDRs) are displayed as dotted (FDR < 0.01) and dashed (FDR < 0.05) lines (n = 3 biological replicates). Statistical significance was determined using limma (Smyth, 2005).

Consistent with a role for eIF4E1b in mRNA storage and eIF4E1c in translation initiation, mRNAs involved in chromatin organization and remodeling have been reported to be deadenylated and stable during the oocyte-to-egg transition in mouse (Lee et al, 2023), whereas mRNAs involved in mRNA processing have been reported to be translated upon fertilization in sea urchin (Chassé et al, 2018). Interestingly, many of the mRNAs enriched in the eIF4E1b RIP were histone mRNAs (Fig. 5A and Dataset EV4), most of which lack a polyA tail and thus cannot be repressed by deadenylation. To investigate the translational status of the mRNAs bound by eIF4E1b and eIF4E1c in zebrafish embryos, we compared our RIP-seq data with publicly available datasets of mRNAs expressed during zebrafish embryonic development. Of note, mRNAs enriched in eIF4E1b RIP were overall less abundant than mRNAs enriched in eIF4E1c RIP in RNA-seq datasets based on polyA-selected and rRNA depletion (RiboMinus) protocols (Cabrera-Quio et al, 2021) (Fig. 5C). Analysis of polyA tail length (Chang et al, 2018) revealed that most transcripts enriched in the eIF4E1b RIP had short polyA tails during the first 4 hpf, in stark contrast to the long polyA tails observed for mRNAs enriched in the eIF4E1c RIP (Fig. 5D). Consistent with mRNAs with long

polyA tails being specifically enriched in eIF4E1c RIP, our IP-MS experiments showed a >300-fold enrichment of the cytoplasmic polyA-binding protein PABPC1L in eIF4E1c IPs from 8-cell embryos, whereas it was <3.5-fold enriched in eIF4E1b IPs (Dataset EV2). Following the general trend of maternal transcripts, the polyA tails of mRNAs enriched in the eIF4E1b RIP at 1.25 hpf also increase in length during embryogenesis (Fig. 5C,D). PolyA tail length is strongly correlated with translation efficiency (TE) during early embryogenesis (Subtelny et al, 2014). Using published TE data (Subtelny et al, 2014), we observed that mRNAs enriched in the eIF4E1b RIP are translationally repressed, whereas mRNAs enriched in the eIF4E1c RIP have high TEs (Fig. 5E) (Subtelny et al, 2014). To directly assess whether translation of an mRNA in the embryo is affected by binding of either canonical eIF4E or eIF4E1b, we tethered eIF4Ea or eIF4E1b to a *GFP* reporter mRNA using the MS2/MCP system (Fig. 5F). Tethering of both eIF4E1b and eIF4Ea caused a decrease in GFP protein levels (Fig. 5G) and a destabilization of the reporter mRNA (Fig. 5H; significant only in the case of eIF4Ea) compared to the untethered control. However, only tethering of eIF4Ea led to a significant increase in translation relative to the amount of reporter mRNA left (Fig. 5I). Similarly,

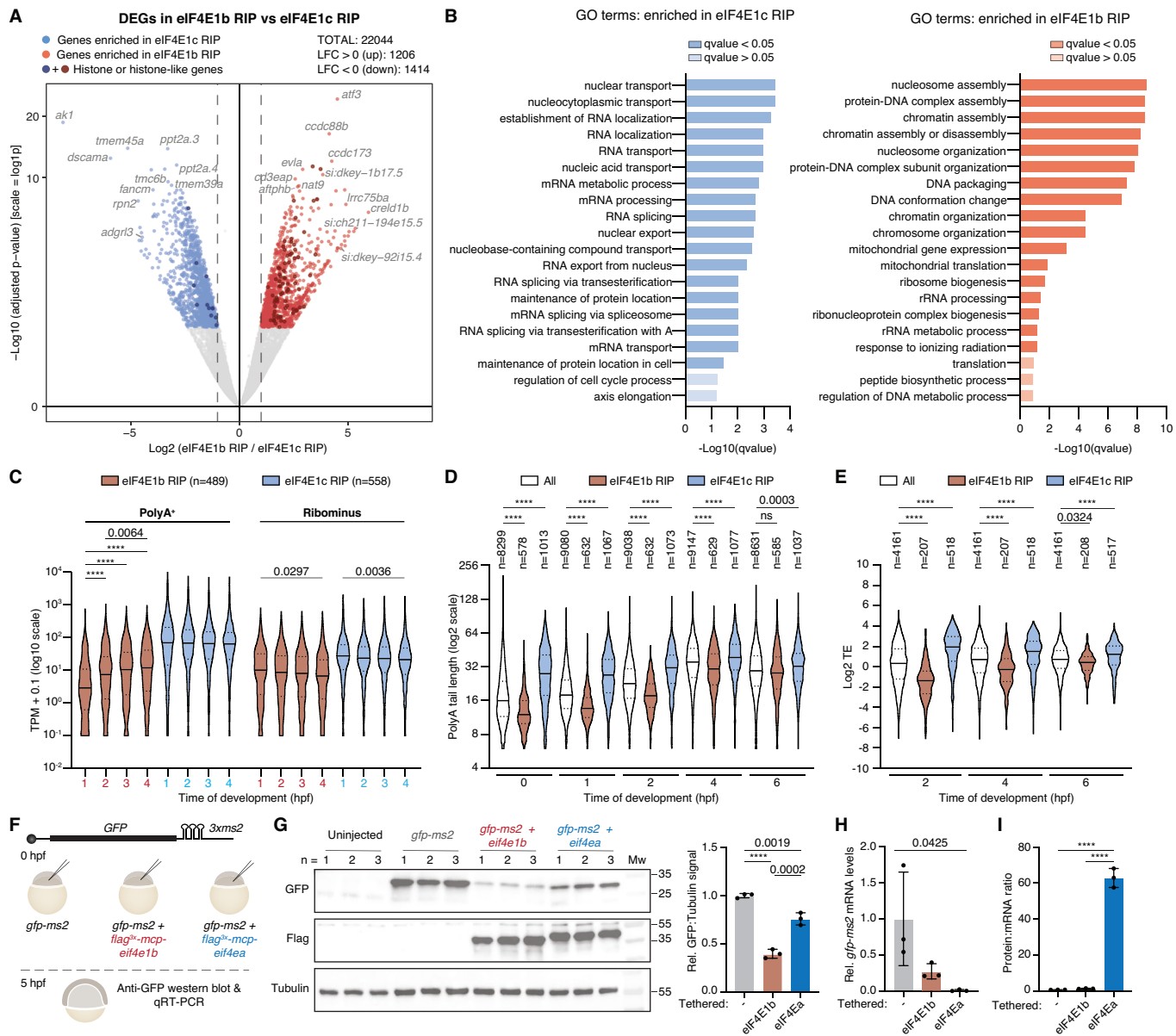

**Figure 5. eIF4E1b binds to translationally repressed mRNAs involved in chromatin regulation.**

(A) Differential expression gene (DEG) analysis of mRNAs immunoprecipitated with eIF4E1b or eIF4E1c at 1.25 h post fertilization. mRNAs significantly enriched in eIF4E1b ($n = 3$ biological replicates) or eIF4E1c ($n = 2$ biological replicates) immunoprecipitations are shown in red and blue, respectively ($P$ value < 0.005). Genes encoding proteins with histone and histone-like domains are highlighted in a darker color. Significance was determined using Benjamini–Hochberg-corrected Wald test (DeSeq2). (B) Gene ontology (GO) analysis of mRNAs that are specifically bound to eIF4E1c (left, in blue) or to eIF4E1b (right, in red). (C) Expression levels (in transcript per million, TPM) of mRNAs enriched in eIF4E1b and eIF4E1c RIPs during zebrafish embryogenesis according to published RNA-seq datasets obtained with polyA-selection (PolyA$^+$) and rRNA depletion (Ribominus) protocols (Cabrera-Quio et al, 2021). (D) PolyA tail length of mRNAs enriched in RIPs with eIF4E1b or eIF4E1c during zebrafish embryonic development based on published TAIL-seq data (Chang et al, 2018). (E) Translational efficiency (TE) of mRNAs enriched in eIF4E1b and eIF4E1c RIPs during zebrafish embryogenesis based on published TE data (Subtelny et al, 2014). (F) Schematic of the eIF4E tethering assay: A *GFP* reporter mRNA containing *MS2* loops at the 3' UTR (top) and an mRNA containing *eIF4Ea* or *eIF4E1b* fused to an N-terminal 3xflag-MCP (MS2 Coat Protein) tag were injected into 1-cell embryos. After 5 h, GFP protein and mRNA levels were assessed by western blot and RT-qPCR, respectively. (G) Western blots showing GFP (top), Flag (middle) and α-Tubulin (loading control; bottom) protein levels in 5 hpf embryos after injection with the mRNAs described in (F). Uninjected embryos are also shown. Quantification of GFP signal normalized to α-Tubulin is shown on the right ($n = 3$ experimental replicates). (H) Levels (assessed by RT-qPCR) of *GFP* reporter mRNA tethered to either eIF4Ea or eIF4E1b compared to non-tethered *GFP* mRNA in 5 hpf embryos ($n = 3$ experimental replicates). (I) Protein to mRNA ratio of the *GFP* reporter mRNA alone or tethered to eIF4Ea or eIF4E1b, as estimated from western blot (G) and RT-qPCR (H) ($n = 3$ experimental replicates). Data information: (C–E) lines indicate median and first and third quartiles; in (G–I), columns indicate mean and error bars show the SD. (C–E) Significance was assessed with Kruskal–Wallis followed by Dunn's test. (G–I) Significance was determined using ordinary one-way ANOVA followed by a Tukey's multiple comparisons test. (C) Significance was calculated for eIF4E1b RIP or eIF4E1c RIP in PolyA$^+$ or Ribominus data (if not stated, $P$ value > 0.005). (D, E) statistics are shown only for comparisons within the same developmental stage. Hpf hours post fertilization. ****$P$ value < 0.0001. Source data are available online for this figure.

tethering of eIF4G to the 3' UTR of an mRNA has been reported to promote translation of the upstream open reading frame (Paek et al, 2015).

Our results above suggest that eIF4E1b does not promote translation. To investigate this further, we analyzed the proteome of *eIF4E1b* and *eIF4E1c* overexpressing (OE) embryos at 5 hpf, a time when eIF4E1b levels in wild type decrease to almost half of their maximum at 2.5 hpf (Fig. 1D). *eIF4E1c* overexpression resulted in mostly increased protein levels (~90% of the dysregulated proteins, 747 proteins, were upregulated), as expected from increasing the levels of a translational activator (Fig. 6A and Dataset EV5). In contrast, and in line with eIF4E1b repressing translation, *eIF4E1b* overexpression resulted in ~76% of the dysregulated proteins being downregulated (Fig. 6B and Dataset EV5). To assess whether the changes in protein levels were also at the mRNA level, we performed RT-qPCR in wild-type, *eIF4E1c* OE and *eIF4E1b* OE embryos, focusing on mRNAs of maternal origin (Bhat et al, 2023). While none of the tested mRNAs encoding proteins upregulated in *eIF4E1c* OE were significantly altered, four out of seven mRNAs encoding proteins upregulated in *eIF4E1b* OE were significantly increased (Fig. 6C), suggesting that the few proteins upregulated in *eIF4E1b* OE embryos are due to an increase in mRNA levels rather than translation. On the other hand, mRNAs encoding proteins that were depleted in *eIF4E1b* OE embryos showed similar mRNA abundances in wild-type and *eIF4E1b* OE embryos (Fig. 6C) and have been reported to be efficiently translated in wild-type embryos (Fig. 6D–F) (Subtelny et al, 2014), suggesting that *eIF4E1b* overexpression represses their translation. We then analyzed the proteome (by MS; Dataset EV6) and transcriptome (by RNA-seq; Dataset EV7) of *eif4e1b* mutant and wild-type female gonads of juvenile fish expressing high levels of *ziwi:GFP* (Appendix Fig. S4A). MS data revealed that ~85% of the 1447 dysregulated proteins were significantly increased in *eif4e1b* mutant gonads (Fig. 6G). The majority (~61%) of the proteins enriched in the mutant gonads were not upregulated at the mRNA level (Fig. 6H,I), in line with an enhanced translation of certain transcripts in the absence of eIF4E1b. Moreover, and consistent with eIF4E1b not promoting translation, ~70% of the proteins depleted in *eif4e1b* mutant gonads were also downregulated at the mRNA level (Fig. 6H,I). Many of the mRNAs depleted in *eif4e1b* mutant gonads were enriched in the eIF4E1b RIP at 1.25 hpf (547 genes, Appendix Fig. S4B–D), suggesting that loss of eIF4E1b destabilizes its mRNA targets. In particular, histone mRNAs were enriched in the eIF4E1b RIP (Fig. 5A and Dataset EV4) and depleted in *eif4e1b* mutant gonads (Fig. 6H and Dataset EV7). While maternal histones are deposited in lipid droplets in Drosophila (Li et al, 2012), similar droplets have so far not been reported in vertebrate embryos. In somatic cells, mRNA synthesis and degradation of so-called replication-dependent (RD) histones are coupled to the G1/S phase of the cell cycle (Graves et al, 1987; Armstrong and Spencer, 2021). The absence of transcription during early embryogenesis requires that RD histone mRNAs, which are highly abundant during the first hours of embryogenesis (Appendix Fig. S4E), are stored in the egg in a translationally repressed state, as translation of RD histone mRNAs is coupled to their decay (Stimac et al, 1983; Tuck et al, 2020). Since RD histone mRNAs lack a polyA tail, we hypothesized that eIF4E1b may play a critical role in their repression and stabilization. Consistent with this, loss of eIF4E1b in zebrafish gonads resulted in a reduction of 141 histone and histone-like

mRNAs, including the canonical core histones *H3*, *H4*, *H2A*, and *H2B* (Fig. 6H and Dataset EV7). In contrast, only 19 histone or histone-like mRNAs were significantly enriched in *eif4e1b* mutant gonads, 9 of which encode zebrafish orthologs of human replication-independent histones that are expressed at all times of the cell cycle and whose mRNAs can be polyadenylated (Marzluff et al, 2008) (Fig. 6H and Dataset EV7).

## Discussion

In the absence of transcription, regulation of gene expression during early development relies on post-transcriptional mechanisms that control maternal mRNA stability and translation. Here we show that eIF4E1b is a germline-specific eIF4E paralog that plays a critical role in female germline development in zebrafish. Importantly, our in vivo and vitro data suggest that, unlike other class I eIF4Es, eIF4E1b does not promote translation because (1) it does not interact with the translational factor eIF4G, (2) it localizes to P-bodies by interacting with the translational repressor eIF4ENIF1, (3) it binds to mRNAs that are reported to have short polyA tails and low TEs in the embryo, (4) it does not promote translation when tethered to an mRNA, and (5) its levels negatively correlate with total protein abundance in vivo. While the importance of eIF4E1b in female germline development is consistent with recent studies in mice showing that eIF4E1B is essential for female fertility (Guo et al, 2023; Yang et al, 2023), the mouse studies propose a role for eIF4E1B in translating specific mRNAs during the oocyte-to-embryo transition (Guo et al, 2023; Yang et al, 2023). Guo et al suggest that eIF4E1B functions as a canonical eIF4E in translation initiation, although they do not provide evidence for an interaction between eIF4E1B and eIF4Gs.

Our RIP-seq data show that eIF4E1b binds to mRNAs reported to have short polyA tails (median 12 nucleotides) and low TEs in the embryo. While shortening polyA tails to 10-12 nucleotides promotes mRNA decapping and degradation in somatic cells (Passmore and Coller, 2022), maternal mRNAs with polyA tails shorter than 12 nucleotides are stable during the first hours of embryogenesis (Fig. EV5F) (Bhat et al, 2023). We propose that eIF4E1b stabilizes mRNAs by binding to the mRNA cap and interfering with the decapping machinery, similar to canonical eIF4E (Schwartz and Parker, 2000; Räsch et al, 2020). Further support for this hypothesis comes from our finding that eIF4E1b has a strong affinity for eIF4ENIF1, a translational repressor that targets eIF4E1b to P-bodies in the embryo. Binding of eIF4ENIF1 to eIF4Es has been shown to stabilize deadenylated mRNAs (Räsch et al, 2020). In Drosophila, the eIF4E-binding protein Cup, ortholog to eIF4ENIF1, has also been reported to target maternal *oskar* mRNA to P-bodies (Bayer et al, 2023) and to be essential for fly oogenesis (Lehmann and Nüsslein-Volhard, 1986; Nakamura et al, 2004). The *C. elegans* eIF4E paralog IFE-3 (Appendix Fig. S5) has also been reported to interact with the worm eIF4ENIF1 ortholog IFET-1 in the germline and to be important for oocyte cell fate (Huggins et al, 2020) and embryonic development (Keiper et al, 2000). We propose that eIF4ENIF1-mediated repression of eIF4E1b-bound maternal mRNAs in P-bodies is also important for vertebrate oogenesis and embryogenesis. Mutations in *eIF4ENIF1* cause infertility in women (Kasippillai et al, 2013; Zhao et al, 2019; Shang et al, 2022), in agreement with both eIF4E1b and eIF4ENIF1 working together to regulate maternal mRNA dormancy.

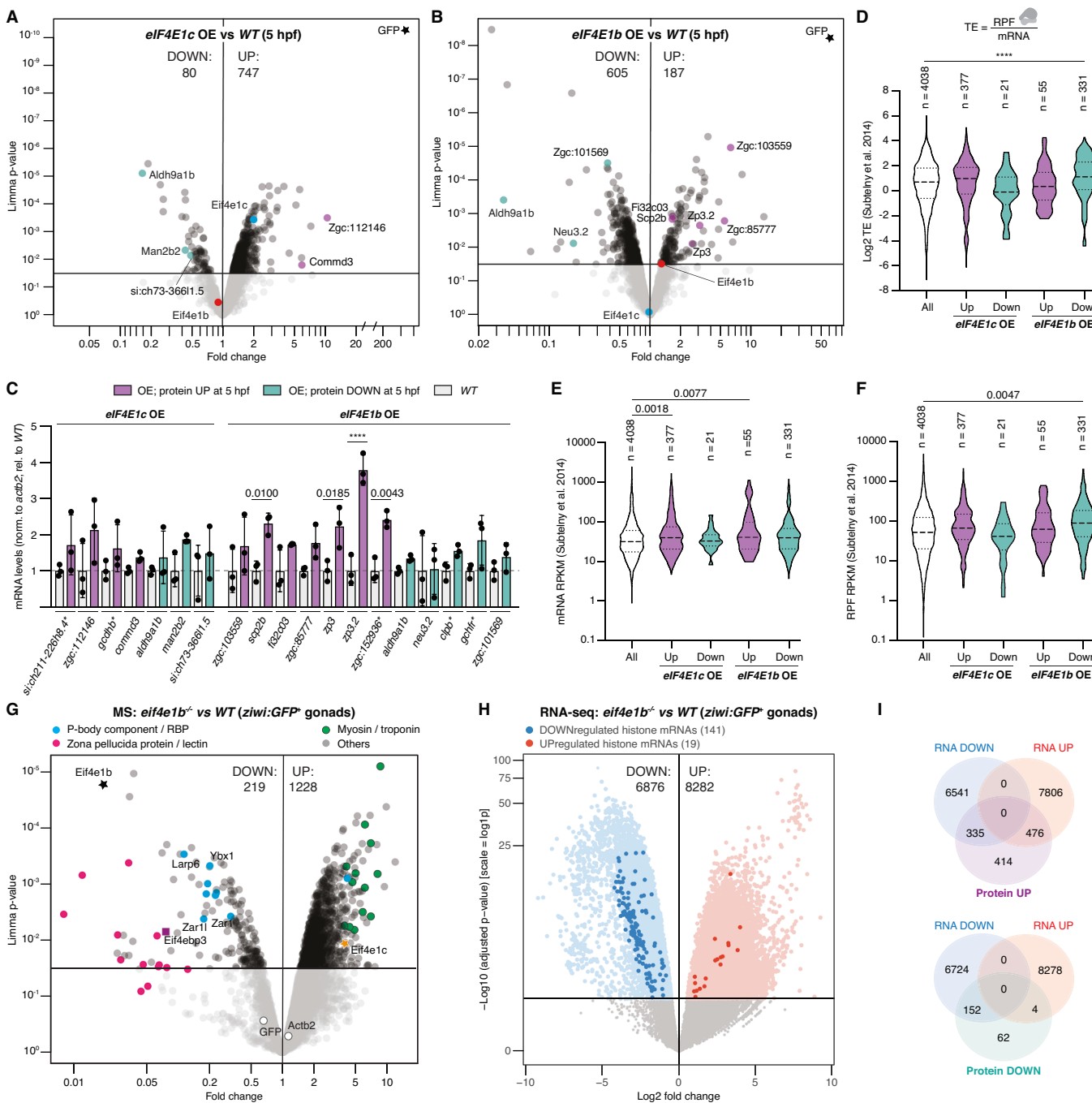

**Figure 6. eIF4E1b levels negatively correlate with total protein levels in zebrafish embryos and gonads.**

(A, B) Volcano plots of MS data obtained from lysates of 5 hpf embryos expressing either *actb2:3xflag-sfGFP-eIF4E1c* (*eIF4E1c* OE; (A)) or *actb2:3xflag-sfGFP-eIF4E1b* (*eIF4E1b* OE; (B)) compared to wild type (*WT*). n = 3 biological replicates. (C) mRNA levels (assessed by RT-qPCR) of proteins dysregulated in *eIF4E1c* OE (left) or *eIF4E1b* OE (right) embryos in *eIF4E1b* OE, *eIF4E1c* OE and *WT* embryos at 5 hpf (n = 3 biological replicates). Columns indicate mean and error bars indicate SD. (D–F) Translational efficiency (TE; (D)), mRNA levels (E) and ribosome-protected fragments (RPF; (F)) of genes encoding proteins dysregulated in *eIF4E1c* OE or *eIF4E1b* OE embryos (data from 4 hpf wild-type embryos from Subtelny et al, 2014). Lines indicate median and first and third quartiles. (G, H) Volcano plots representing mass spectrometry (MS; (G)) and RNA sequencing (RNA-seq; (H)) data from *eif4e1b* mutant and wild-type female gonads isolated from juvenile fish with high *ziwi:GFP* expression (n = 3 biological replicates). (I) Venn diagram showing the overlap between up- or downregulated proteins with up- or downregulated mRNAs in *eif4e1b* mutant gonads isolated from high *ziwi:GFP*-expressing juvenile fish. Data information: significance was calculated using an ordinary one-way ANOVA followed by Sidak's (C), Dunett's multiple comparisons tests (D–F), limma (A, B, G, Smyth, 2005), and Benjamini–Hochberg-corrected Wald test (DeSeq2; H). ****P value < 0.001. RPKM reads per kilobase of transcript per million mapped reads, Hpf hours post fertilization. Source data are available online for this figure.

Interestingly, loss of eIF4E1b in the female gonads of juvenile fish results in increased eIF4E1c levels (Fig. 6G and Dataset EV6), which may partially compensate for the reduction in mRNA stability caused by eIF4E1b not bound to the mRNA cap. Since eIF4E has been implicated in cancer (Hsieh and Ruggero, 2010), the upregulation of eIF4E1c may contribute to the rare development of tumor-like gonads without oocytes in *eif4e1b* mutant fish. Consistent with this phenotype, genes involved in cell adhesion, such as *cdh2* and *cd44a*, which have been implicated in ovarian cancer (Zhang et al, 2008; Kim et al, 2012), were upregulated at the mRNA level in *eif4e1b* mutant gonads (Appendix Fig. S4B,F and Dataset EV7).

A key open question is what determines the selectivity of eIF4E1b for certain mRNAs. Since eIF4E1b binds to the mRNA cap, one possibility is that the specificity is determined by RNA-binding proteins. Ybx1, Zar1, Zar1l, Slbp2, Larp6a, and Larp6b immunoprecipitated with eIF4E1b in oocytes and/or embryos and were less abundant in *eif4e1b* mutant gonads (see Datasets EV1, EV2, and EV6), suggesting that they may function together with eIF4E1b. In zebrafish, *larp6a*, *larp6b* mutant eggs were reported to contain reduced levels of zona pellucida and lectin-type proteins (Hau et al, 2020), a phenotype we also observed in the gonads of juvenile fish lacking eIF4E1b (Fig. 6G; Dataset EV6). Moreover, Larp6 acts in concert with myosins (Wang and Stefanovic, 2014), which we found enriched in *eif4e1b* mutant gonads (Fig. 6G; Dataset EV6). While these proteins may determine the specificity of eIF4E1b for certain mRNAs, understanding their contribution to the regulation of maternal mRNA dormancy will require future work. Another interesting open question is whether eIF4E1b remains bound to its target mRNAs during cytoplasmic poly-adenylation. A release of transcripts from eIF4E1b during early embryogenesis in mice has been suggested by Yang et al, yet future studies will be needed to elucidate the mechanism. The conservation of eIF4E1b in vertebrates and its expression in the human brain (Human Protein Atlas version 22.0) suggest that our findings from the zebrafish oocyte-to-embryo transition may have direct relevance in other systems and cellular contexts.

## Methods

### Zebrafish husbandry

Zebrafish (*Danio rerio*) were raised at 28 °C with a 14/10 h of light/dark cycle. Wild-type TLAB fish correspond to fish obtained by crossing AB with the natural variant TL (TupfelLongfin). All fish experiments were conducted according to Austrian and European guidelines for animal research and approved by local Austrian authorities (protocols for work GZ342445/2016/12 and MA 58-221180-2021-16). Zebrafish *eif4e1b* mutants and *actb2:3xflag-sfGFP-eIF4E1b*, *actb2:3xflag-sfGFP-eIF4E1c*, *actb2:3xflag-sfGFP-eIF4Ea* and *ziwi:GFP* transgenic lines were generated in this study and are described below.

### Zebrafish knockout and transgenic lines

Zebrafish *eif4e1b* knockout fish were generated by CRISPR-Cas9 mediated mutagenesis. Guide-RNAs (gRNAs) targeting the third exon of *eIF4E1b* (5'-CACCAAATTCGACACGGTCGAGG-3' and

5'-GACACGGTCGAGGACTTCTGGGG-3') were injected together with recombinant Cas9 protein (Molecular Biology Service, IMP) into 1-cell zebrafish embryos. To identify fish carrying mutations in the germline (i.e., founders), adult fish were crossed to wild type; embryos were genotyped by PCR using 5'-AGATGGG-GACTTTGGTTCTACA-3' and 5'-TGCTCTACTCCACCTTTCA-CAA-3' primers. Embryos showing a size difference in PCR amplicons were raised to adulthood and heterozygous fish were identified by PCR-based genotyping of fin clips. Homozygous fish and their wild-type siblings were generated by crossing hetero-zygous fish. Sex was determined at 4–5 months post fertilization based on the sexual dimorphism characteristic of zebrafish. Pictures of anesthetized fish (in MESAB) were taken on a ZEISS Stemi 508 stereomicroscope with camera (×2 magnification, FlyCapture2 software). Fertilization rates were counted at 3 hpf; unfertilized eggs were removed, and embryo survival was counted at 1 dpf.

To generate transgenic lines, the coding sequences of full-length eIF4Ea, eIF4E1b and eIF4E1c were PCR-amplified from zebrafish cDNA and cloned by Gibson assembly into a vector containing Tol2-integrations sites in between the zebrafish *actb2* promoter, *actb2* 5' UTR, an N-terminal *3xflag-sfGFP* tag and the SV40 late polyadenylation signal (*SVLPA*). In total, 15 pg of each plasmid was co-injected with 35 pg of *Tol2* mRNA into 1-cell embryos. Adult fish were crossed with wild-type fish, and GFP-positive embryos were raised to adulthood. Rescue lines were obtained by injecting the *3xflag-sfGFP-eIF4E1b* vector into *eif4e1b* heterozygous embryos. Homozygous fish and wild-type fish were obtained by crossing *eif4e1b* heterozygous embryos expressing 3xflag-sfGFP-eIF4E1b.

We also generated *ziwi:EGFP* transgenic fish by injecting a plasmid containing the *ziwi* promoter sequence driving *EGFP* (Leu and Draper, 2010). Juvenile fish expressing GFP in the gonads were selected as founders and crossed with *eif4e1b* homozygous fish. Wild-type and homozygous *eif4e1b* siblings containing the *ziwi:EGFP* transgene were further obtained by incrossing and genotyping.

### Phylogenetic analysis

Sequences were collected from the UniProt or NCBI protein databases with NCBI blast+ (Camacho et al, 2009) and aligned with mafft (v7.505, -linsi method) (Katoh and Toh, 2008). For phylogenetic analysis, a maximum likelihood tree was inferred with iqtree2 v.2.2.0 (Minh et al, 2020), with standard model selection using ModelFinder (Kalyaanamoorthy et al, 2017) and ultrafast bootstrap (UFBoot2) support values (Hoang et al, 2018). The tree was visualized in iTOL v6 (Letunic and Bork, 2021).

### Tandem mass tag mass spectrometry (TMT-MS)

#### Sample preparation

Embryos were manually dechorionated and deyolked in Danieau's buffer, containing 58 mM NaCl, 0.7 mM KCl, 0.4 mM MgSO$_4$, 0.6 mM Ca(NO$_3$)$_2$ and 5 mM HEPES pH 7.2, and snap-frozen in liquid nitrogen. In all, 64 embryos per stage from 16 to 19 females were shaken at 800 rpm for 5 min at 95 °C in 4% SDS, 0.1 MDTT, 0.1 M Tris-HCl pH 7.5, and 300 pmol of 7 recombinantly expressed proteins (dCas9, SpoIVB deltaN34 from *G. stearothermophilus*,

ClpC NTD from *B. subtilis*, lambda exonuclease, YwlE from *G. stearothermophilus*, McsA from *S. aureus* and SpoIVFAGs.deltaN137_L234M; Molecular Biology Service, IMP). Lysates were sonicated and the SDS was removed using filter-aided sample preparation (FASP) (Wiśniewski et al, 2009). Proteins were digested using Trypsin Gold (Promega) in 100 mM HEPES pH 7.6 at 37 °C overnight. Samples were acidified with 10% TFA and purified using a Sep-Pak C18 Vac cartridge (Waters). Peptides were dissolved in 100 mM HEPES pH 7.6. Equal volumes of lysates were incubated with TMT10plex reagents for 2 h. After adding 5% hydroxylamine, samples were purified using a Sep-Pak C18 Vac cartridge (Waters), dried, and dissolved in SCX buffer A (5 mM phosphate buffer pH 2.7, 15% ACN).

### LC-MS

MS analysis was performed on an UltiMate 3000 RSLC nano system (ThermoFisher) coupled to a Q Exactive HF-X mass spectrometer (ThermoFisher) with a Proxeon nanospray ion source (ThermoFisher) as described in Leesch et al, 2023. The mass spectrometer was operated in data-dependent mode, with a full scan ($m/z$ range 380–1650, resolution of 120,000, target value 3E6), followed by MS/MS scans of the ten most abundant ions. MS/MS spectra were acquired using a normalized collision energy of 35, isolation width of 0.7 $m/z$, resolution of 45,000, a target value of 1E5, and maximum fill time of 250 ms. For the detection of the TMT reporter ions, a fixed first mass of 110 $m/z$ was set. Precursor ions selected for fragmentation (excluding ions with charge state 1, 7, 8, >8) were put on an exclusion list for 30 sec. The minimum AGC target was set to 1E4 and intensity threshold was calculated to be 4E4.

### TMT-MS data analysis

Peptide spectra were identified with Proteome Discoverer (v2.3.0.523, ThermoFisher) and searched against a custom-made protein database covering GRCz10, GRCz11, RefSeq, UniProt and PDB identifiers (58,522 sequences; 34,078,760 residues), using MS Amanda (v2.0.0.14114) (Dorfer et al, 2014) with the following parameters: iodoacetamide derivative on cysteine was set as a fixed modification, whereas oxidation on Met, deamidation on Asn and Gln, phosphorylation on Ser, Thr, and Tyr and TMT10plex on Lys and peptide N-termini were set as variable modifications. Monoisotopic masses were searched within unrestricted protein masses for tryptic enzymatic specificity. The peptide mass tolerance was set to ± 5 parts per million (ppm) and the fragment mass tolerance to ± 15 ppm. The maximal number of missed cleavages was set to 2. Results were filtered to 1% false discovery rate (FDR) on protein level using the Percolator algorithm integrated in Proteome Discoverer (Käll et al, 2007). Peptides were quantified based on TMT reporter ion intensities using the Reporter Ion Quantifier Node in Proteome Discoverer.

### Histology of adult zebrafish ovaries

Ovaries from 7-month-old wild-type and *eif4e1b* knockout females (three fish per genotype) were dissected under the scope and fixed in 3.7% paraformaldehyde (PFA) in PBS overnight at 4 °C. Embryos were washed in PBS and embedded in 2% agarose followed by dehydration and paraffin infiltration in an automated tissue processor (Donatello Series 1, DiaPath). The processed

agarose blocks were then embedded in paraffin using an embedding station (Tissue-Tek TEC, Sakura), and sectioned at 2 µm on a microtome Microm HM 355 S Leica (ThermoFisher). The sections were dried at 50 °C overnight. Prior to staining procedures, the slides were dewaxed in Thermo Scientific Shandon Xylene Substitute and further rehydrated through decreasing ethanol series using an automatic stainer (Epredia Gemini AS). Hematoxylin and eosin (H&E) staining was also performed using Epredia Gemini AS. For phospho-histone H3 (phH3) staining, the slides were incubated for 30 min at 100 °C in EDTA retrieval solution. After cooling, slides were washed in TBS and incubated in the dark for 10 min with 3% $H_2O_2$. Slides were then washed with TBS and transferred to TBST. After creating a hydrophobic barrier with a PAP Pen (Abcam), the slides were incubated with 5% BSA in TBST supplemented with 10% goat serum (Merck) for 1 h at room temperature (RT). Next, the slides were stained for 1 h with phH3 (06-570, Merck) at RT. After washing 3 times with TBST, slides were incubated with the polymer rabbit detection system (DCS) and phH3 staining was detected after incubating the slides with DAB substrate kit (Abcam) for 10 min at RT. The slides were counterstained with Epredia Shandon Harris Hematoxylin (Fisher Scientific) and further dehydrated using the Epredia Gemini AS. All slides were air-dried overnight and covered with Eukitt Neo medium for coverslipper (O. Kindler) using an automatic coverslipper (Tissue-TEK GLC, Sakura). Images were taken with a Slide Scanner Pannoramic 250 (software version 3.0.2.127553; scanner hardware ID P250F20J2101).

## Protein expression in *E. coli*

The coding sequences of eIF4Ea (34–215; Uniprot: A8E579), eIF4E1c (49–230; Uniprot: B8A6A1), eIF4E2 (52–236; Uniprot: B2GPF6), and eIF4E3 (44–224; Uniprot: Q66HY7), were amplified from cDNA obtained from zebrafish embryos. The coding sequences of mouse (63–244; Uniprot: Q3UTA9) and human (60–242; Uniprot: A6NMX2) eIF4E1B were obtained as synthetic genes from Twist Bioscience. eIF4E coding sequences were cloned into a pOPINB vector providing an N-terminal 10xHis tag followed by an MBP tag and a 3C protease cleavage site. Cloning was done via Gibson assembly and specific mutations and deletions were introduced via site-directed mutagenesis. Plasmids containing the eIF4E-binding motifs of human eIF4G (608-647), eIF4EBP1 (50-83) and eIF4ENIF1 (27-63) with N-terminal MBP and C-terminal GB1 tags were obtained from C Igreja (MPI for Biology, Tübingen) (eIF4G and eIF4EBP1 plasmids were previously published in Peter et al, 2015; Grüner et al, 2016). Plasmids were transformed in *E. coli* BL21 (DE3) cells. For protein expression, cells were grown in LB medium supplemented with antibiotic at 37 °C until reaching an $OD_{600}$ in between 0.6 and 1. Cultures were induced with 0.25 mM IPTG (ThermoFisher) and grown overnight at 18 °C. Pellets were collected by centrifugation at 3900×*g* for 20 min.

## Pulldowns

Bacteria pellets were resuspended in cold lysis buffer containing 50 mM Tris-HCl pH 7.5, 100 mM NaCl, 1 mM DTT (Merck), 10 µg/mL DNase I (Merck) and cOmplete, EDTA-free protease inhibitor cocktail (Merck). Cells were lysed by sonication and lysates were collected via centrifugation at 21,000×*g* for 20 min.

To test for protein–protein interactions, the supernatants (soluble fractions) were supplemented with imidazole (Merck) to reach a final concentration of 20 mM. HisPur Ni-NTA magnetic beads, previously equilibrated with washing buffer containing 50 mM Tris-HCl pH 7.5, 100 mM NaCl, 1 mM DTT (Merck), 20 mM imidazole (Merck) and 0.01% Tween-20 (Merck), were incubated for 1 h with either the soluble fraction of the lysates containing His-tagged proteins (eIF4E or Lsm14[5-79]) or with lysis buffer at 4 °C. After 1 h, lysates were discarded, and beads were incubated with the soluble fraction of the putative interactors for 1 h at 4 °C. Beads were washed three times with washing buffer and bound proteins were eluted by incubating with the beads with elution buffer, containing 50 mM Tris-HCl pH 7.5, 100 mM NaCl, 1 mM DTT (Merck) and 500 mM imidazole (Merck), for 15 min at RT. The elution step was performed twice, and eluates were combined in one tube.

To test for mRNA cap binding, γ-aminophenyl-m$^7$GTP (C10-spacer) agarose beads (Jena Bioscience) were packed into Pierce Screw Cap Spin Columns (ThermoFisher) and equilibrated with washing buffer containing 50 mM Tris-HCl pH 7.5, 100 mM NaCl, 1 mM DTT (Merck) and 0.01% Tween-20 (Merck). The soluble fractions of *E. coli* lysates were added to the columns and incubated for 5 min at RT before centrifugation at 2000×*g* for 30 s. After repeating this step twice, beads were washed three times with washing buffer and incubated for 10 min with 2× Laemmli-sample buffer (Bio-Rad). The elution step was performed twice, and eluates were combined in one tube.

Boiled samples were analyzed by SDS-PAGE using Any kD precast Polyacrylamide Gels (Bio-Rad). Gels were stained with InstantBlue Coomassie Protein Stain (Abcam). Raw images of the gels were quantified with Fiji.

## AlphaFold predictions and structural analysis

Structural predictions of protein complexes were performed with ColabFold (for the generation of multiple sequence alignments) and AlphaFold-Multimer (for the generation of structures) (Mirdita et al, 2022; Evans et al, 2022). Structures were visualized with ChimeraX-1.2.5 or Chimera 1.13.1. Models were aligned using the command *mmaker*.

## mRNA injections into zebrafish eggs

The coding sequences of full-length *Dcp2* (Uniprot: Q6NYI8), *Ddx6* (Uniprot: E7FD91), *Ybx1* (Uniprot: B5DE31-2), *eIF4E1b* (Uniprot: Q9PW28), and *eIF4Ea* (see above) were PCR-amplified from zebrafish cDNA and cloned via Gibson assembly into a vector containing SP6 and T3 promoters, the 5' and 3' UTRs of Xenopus *β-globin* and an A$^{29}$, C$^{14}$ tail. P-body markers contained N-terminal *3xflag* and *dsRed* tags spaced by linkers, whereas *eIF4E1b* and *eIF4Ea* were tagged with N-terminal *3xflag* and *sfGFP* tags spaced by linkers. Specific mutations in *eIF4E1b* were introduced via site-directed mutagenesis. Capped mRNAs were transcribed from linearized plasmids using SP6 or T3 mMessanger Machine Kit (Ambion), according to the manufacturer's protocol. mRNAs were purified with the RNA Clean & Concentrator kit (Zymo Research). For all injections, 100 pg of mRNAs were injected into one-cell stage embryos. Nuclei were labeled with *histone 2B (H2B)-RFP* mRNA (Stock et al, 2022).

## Microscopy

For confocal imaging, oocytes or manually dechorionated embryos were mounted in a drop of 0.8% low-melting agarose in PBS on round glass bottom dishes (Ibidi). For determining the localization of eIF4E proteins at specific stages of oogenesis and embryogenesis, samples were fixed in 3.7% paraformaldehyde (PFA) in PBS overnight at 4 °C. Samples were then washed in PBS and permeabilized with PBS-0.5% Triton X-100 for 30 min at RT; for nuclei or mitochondria staining, samples were incubated with DAPI (Sigma; 1:5000 dilution, 15 min incubation at RT) or Mitotracker Deep Red FM (Invitrogen; 1:2000 dilution, 30 min incubation at RT), respectively. Dishes were filled with E3 medium (5 mM NaCl, 0.17 mM KCl, 0.33 mM CaCl$_2$, 0.33 mM MgSO$_4$, 10$^{-5}$% methylene blue). Images were acquired on an inverted LSM800 Axio Observer (Zeiss) with a temperature of incubation of 27 °C for live imaging. The following objectives were used: Plan-Neofluor 10×/0.3 (Zeiss), Plan-Apochromat 20×/0.8 M27 (Zeiss) or Plan-Apochromat 63×/1.2 water objective (Zeiss). Images from different fluorescent channels were acquired separately with a GaAsP-Pmt detector using different excitation (ex) and detection (em) wavelengths depending on the fluorophore used (ex/em): dsRed (561/563–617 nm), sfGFP (488/400–527, 502–700 or 410–521 nm depending on the experiment), mRFP1 (561/564–617 nm), farRed (640/656-700 nm) or DAPI (405/410–456 nm). ZenBlue 3.1 (Zeiss) was used for image acquisition. For granule quantification of mRNA injections, 2D-images were taken with the Plan-Apochromat 63×/1.2 water objective (Zeiss). For detecting and quantifying the granules, we manually annotated a training dataset and created a custom TensorFlow model using CSBDeep/CARE, which we then applied to our images via Fiji. A threshold was set on the resulting images for subsequent segmentation and the resulting regions were counted and measured.

For fluorescent widefield microscopy, juvenile fish of 1–2 cm in length were anesthetized in 0.1% (w/v) tricaine (Sigma). Images were acquired on a fluorescent stereomicroscope (Lumar.V12, Zeiss) with a NeoLumar S 0.8x objective (Zeiss), an Axiocam 702mono (Zeiss), and an X-cite Xylis (Exelitas) fluorescent lamp using ZenBlue 3.1 (Zeiss) software.

Images were post-processed with ImageJ 1.53t software adjusting brightness and contrast.

## Immunoprecipitation followed by mass spectrometry (IP-MS)

Embryos were dechorionated with 1 mg/mL of pronase (Sigma-Aldrich), lysed in cold lysis buffer containing 100 mM Tris-HCL pH 7.5, 150 mM NaCl, 0.05% Triton X-100 (Sigma-Aldrich) and cOmplete, EDTA-free protease inhibitor cocktail (Merck) and homogenized with a Dounce homogenizer (Sigma-Aldrich). Lysates were cleared at 13,000×*g* for 10 min at 4 °C and incubated with anti-GFP agarose beads (Molecular Biology Service, IMP), which were previously equilibrated with lysis buffer overnight at 4 °C. Beads were then washed ten times with washing buffer containing 100 mM Tris-HCl pH 7.5 and 150 mM NaCl. For RNAse treatment, lysates were incubated with 1 unit of RNAse I (Ambion) per 500 µL lysate for 30 min at 37 °C. For IP on squeezed eggs, mature oocytes were squeezed from two females into Petri dishes and immediately

lysed and homogenized. For IP on oocytes, ovaries from three female fish were pooled. Early (stages I and II) and late (stages III and IV) oocytes were harvested according to Jamieson-Lucy and Mullins, 2019b with some modifications. Briefly, the pooled ovaries were kept in Leibovitz's L-15 medium (Gibco) at 28 °C. Ovaries were incubated in digestive mix containing 1 mg/ml collagenase I (Sigma-Aldrich), 1 mg/ml collagenase II (Sigma-Aldrich), and 0.5 mg/ml hyaluronidase IV (Sigma-Aldrich) in Leibovitz's L-15 medium for 30 min at RT on a tube rotator. After stopping the digest with Leibovitz's L-15 medium, oocytes were manually dissociated, washed with Leibovitz's L-15 medium, and lysed and homogenized as described before.

HPLC-MS was performed using an UltiMate 3000 RSLC nano system coupled to an Orbitrap Exploris 480 mass spectrometer, equipped with a Proxeon nanospray source or to an Orbitrap Eclipse Tribrid mass spectrometer equipped with a FAIMS pro interface and a Nanospray Flex ion source (all parts ThermoFisher Scientific). Peptides were loaded onto a trap column (ThermoFisher Scientific, PepMap C18, 5 mm × 300 μm ID, 5-μm particles, 100 Å pore size) at a flow rate of 25 μL/min using 0.1% TFA as mobile phase. After 10 min, the trap column was switched in line with the analytical column (ThermoFisher Scientific, PepMap C18, 500 mm × 75 μm ID, 2 μm, 100 Å). Peptides were eluted using a flow rate of 230 nl/min, starting with the mobile phases 98% A (0.1% formic acid in water) and 2% B (80% acetonitrile, 0.1% formic acid) and linearly increasing to 35% B over the next 120 min, followed by a gradient to 95% B in 5 min, staying there for 5 min and decreasing in 2 min back to the gradient 98% A and 2% B for equilibration at 30 °C. The Orbitrap Exploris 480 mass spectrometer was operated in data-dependent mode, performing a full scan ($m/z$ range 350–1200, resolution 60,000, normalized AGC target 100%) at three different compensation voltages (CV −45, −60, −75), followed each by MS/MS scans of the most abundant ions for a cycle time of 0.9 (CV −45, −60) or 0.7 (CV −75) seconds per CV. MS/MS spectra were acquired using HCD collision energy of 30, isolation width of 1.0 $m/z$, orbitrap resolution of 30,000, normalized AGC target 200%, minimum intensity of 25,000 and maximum injection time of 100 ms. Precursor ions selected for fragmentation (include charge state 2–6) were excluded for 45 s. The monoisotopic precursor selection (MIPS) filter and exclude isotopes feature were enabled. The Eclipse was operated in data-dependent mode, performing a full scan ($m/z$ range 375–1500, resolution 60k, AGC target value 400,000, normalized AGC target 100%) at three different compensation voltages (CV −45, −60, −75), followed each by MS/MS scans of the most abundant ions for a cycle time of 0.9 s per CV. MS/MS spectra were acquired using an isolation width of 1.0 $m/z$, Orbitrap resolution of 30,000, AGC target value of 100,000, intensity threshold of 25,000 and a maximum injection time of 50 ms, using the Orbitrap for detection with HCD collision energy of 30. Precursor ions selected for fragmentation (include charge state 2–6) were excluded for 45 s. The monoisotopic precursor selection filter and exclude isotopes feature were enabled.

Raw MS data from IP experiments was loaded into Proteome Discoverer (PD, version 2.5.0.400, Thermo Scientific). All MS/MS spectra were searched using MS Amanda v2.0.0.19924 (Dorfer et al, 2014). Trypsin was specified as a proteolytic enzyme cleaving after lysine and arginine (K and R) without proline restriction, allowing for up to two missed cleavages. Mass tolerances were set to ±10 ppm at the precursor and ±10 ppm at the fragment mass level, increasing to ±20 ppm for the Eclipse data. Peptide and protein identification was performed in two steps. An initial search was performed against the ENSEMBL database, using taxonomy zebrafish (release 2021_06; 45,694 sequences; 26,805,620 residues), with common contaminants appended. Here, beta-methylthiolation of cysteine was searched as fixed modification, whereas oxidation of methionine, deamidation of asparagine and glutamine and glutamine to pyro-glutamate conversion at peptide N-termini were defined as variable modifications. Results were filtered for a minimum peptide length of seven amino acids and 1% FDR at the peptide spectrum match (PSM) and the protein level using the Percolator algorithm (Käll et al, 2007) integrated in Proteome Discoverer. In addition, an Amanda score of at least 150 was required. Identified proteins were exported and subjected to a second-step search considering phosphorylation of serines, threonines, and tyrosines as additional variable modifications. The localization of the post-translational modification sites within the peptides was performed with the tool ptmRS, based on the tool phosphoRS (Taus et al, 2011). Identifications were filtered using the filtering criteria described above, including an additional minimum PSM-count per protein in at least one sample of 2. The identifications were subjected to label-free quantification using IMP-apQuant (Doblmann et al, 2019). Proteins were quantified by summing unique and razor peptides and applying intensity-based absolute quantification (iBAQ) (Schwanhäusser et al, 2011). Protein abundance normalization was done using sum normalization. The statistical significance of differentially expressed proteins was determined using limma (Smyth, 2005).

## Mass spectrometry (MS) on zebrafish gonads and embryos

Juvenile transgenic fish (1–1.4 cm long) expressing *ziwi:GFP* in the gonads were imaged on a Zeiss Lumar.V12 wildefield stereomicroscope coupled to an LED fluorescence light source (Illumination: X-Cite Xylis; Excelitas technologies) and a Zeiss Axiocam 702 Mono camera. Fish with high GFP expression were then fin-clipped and genotyped for the *eif4e1b* mutation. Gonads from *ziwi:GFP* transgenic wild-type and *eif4e1b* mutant juveniles expressing high levels of GFP were dissected under a scope. Five gonads per sample were collected and snap-frozen in liquid nitrogen.

In all, 5 hpf embryos (100 embryos per replicate) were dechorionated with 1 mg/mL of pronase and deyolked by adding 1 mL of cold deyolking buffer (55 mM NaCl, 1.8 mM KCl and 1.25 mM NaHCO$_3$). Embryos were shaken at 1100 rpm for 5 min at 4 °C. Cells were then pelleted at 300×g for 30 s and snap-frozen.

Samples were mixed with 300 μL of methanol and sonicated for 30 cycles (0.5 s, 50% amplitude) using an ultrasonication probe connected to the Ultrasonic processor UP100H (Hielscher). Then, 500 μL of chloroform was added to each vial and, after 5 min, proteins were collected by centrifugation at 10,000 rcf for 5 min. After air drying, protein pellets were resuspended in 72.5 μL of 10 M urea (in 50 mM HCl) and incubated at RT for 10 min. Then, 7.5 μL of 1 M TEAB pH 8 was added and protein levels were determined by Bradford. After the addition of 1 μl benzonase and 1.25 μL 1 M DTT, samples were incubated at 37 °C for 1 h. Alkylation was performed by incubating the samples with 2.5 μL of 1 M IAA for 30 min in the dark. The reactions were quenched

with 0.6 μL of 1 M DTT. Samples were diluted with 100 mM TEAB buffer pH 8.5 to a final urea concentration of 6 M and digested with LysC at a 1:50 enzyme:protein ratio for 3 h at 37 °C. After the predigestion with LysC (Wako), samples were diluted with 100 mM TEAB pH 8.5 to a final urea concentration of 2 M. Tryptic digestion (Trypsin Gold, Promega) was at a 1:50 enzyme:protein ratio for overnight at 37 °C.

As before, HPLC-MS was performed using an UltiMate 3000 RSLC nano system coupled to an Orbitrap Exploris 480 mass spectrometer, equipped with a FAIMS pro interface and a Nanospray Flex ion source (all parts ThermoFisher Scientific). 750 ng of peptides were loaded onto a trap column (PepMap Acclaim C18, 5 mm × 300 μm ID, 5-μm particles, 100 Å pore size, ThermoFisher Scientific) at a flow rate of 25 μl/min using 0.1% TFA as mobile phase. Further steps were performed as described above, except that MS/MS spectra were acquired using a collision energy of 30, an isolation width of 1.2 $m/z$, a resolution of 15,000, a maximum fill time of 30 ms, a normalized AGC target of 1E5 and an intensity threshold of 2.5E4. Precursor ions selected for fragmentation (including charge states 2–6) were excluded for 40 s.

## Western blotting

Lysates from early oocytes (stages I–II) were supplemented with 4× Laemmle-sample buffer (Bio-Rad). 4 hpf embryos were deyolked in 55 mM NaCl, 1.8 mM KCl, 1.25 mM NaHCO$_3$, and cOmplete, EDTA-free protease inhibitor cocktail (Merck). Embryos were lysed by pipetting and the yolk was dissociated by shaking the lysates at 1100 rpm for 5 min at 4 °C. Cells were pelleted by centrifugation at 500 rpm for 30 s and resuspended in 4× Laemmle-sample buffer (Bio-Rad). Oocyte and embryo samples were boiled at 95 °C for 5 min and separated by SDS-PAGE using Any kD precast Polyacrylamide Gels (Bio-Rad). Blotting was performed onto a nitrocellulose membrane (GE Healthcare) using a Bio-Rad wet blot system. Membranes were blocked for 1 h at RT with 3% BSA in TBST, and the following antibodies were added to a solution containing 1.5% BSA in TBST: anti-Dcp2 (rabbit, 1:1000, NBP2-16109, Novus Biological), which was previously tested in zebrafish embryos (Zampedri et al, 2016), anti-Rps17 (rabbit, 1:1000, ab128671, Abcam), anti-α-tubulin (mouse, 1:20,000, T6074, Sigma-Aldrich), anti-GFP (rabbit, 1:1000, A11122, Invitrogen), or anti-Flag (mouse, 1:1000, F1804, Sigma-Aldrich). After washing with TBST, membranes were incubated with F(ab′)2 anti-rabbit IgG (H + L)-HRPO (goat, 1:20,000, 111-036-045, Dianova) or F(ab′)2 anti-mouse IgG (H + L)-HRPO (goat, 1:10,000, 115-036-062, Dianova) secondary antibodies. Chemiluminescence was induced by using Clarity Western ECL Substrate (Bio-Rad).

## RNA immunoprecipitation (RIP)

In all, 8-cell or 1k-cell stage embryos were dechorionated with 1 mg/mL of pronase. RIP was performed as described in Ren et al, 2020 with some modifications. Briefly, embryos were lysed in cold RIP buffer containing 50 mM Tris-HCl pH 7.5, 150 mM NaCl, 5 mM MgCl$_2$, 0.5% IGEPAL® CA-630 (Sigma-Aldrich), 1 mM DTT (Merck), cOmplete, EDTA-free protease inhibitor cocktail (Merck) and 40 U/μL RNAse Out (Invitrogen) and homogenized with a Dounce homogenizer (Sigma-Aldrich). Lysates were cleared at 13,000×$g$ for 10 min at 4 °C and incubated with anti-GFP agarose

beads (Molecular Biology Service, IMP), previously equilibrated with RIP buffer for 2 h at 4 °C. Overall, 200 μL of each lysate, corresponding to the input sample, was kept on ice. Beads were washed two times with RIP buffer and two times with washing buffer containing 50 mM Tris-HCl pH 7.5, 150 mM NaCl, 5 mM MgCl$_2$, and 0.5% IGEPAL® CA-630 (Sigma-Aldrich). Beads were incubated with 0.3 μg/μl proteinase K, 50 mM Tris-HCL pH 7.5, 150 mM NaCl, 5 mM MgCl$_2$, 0.5% IGEPAL® CA-630 (Sigma-Aldrich) and 1% SDS for 30 min at 55 °C. RNA from elution and input samples was purified with the RNA Clean & Concentrator kit (Zymo Research) according to the manufacturer's protocol.

## Quantitative RT-PCR (qPCR)

RNA from zebrafish embryos was isolated using the RNeasy Mini Kit (Qiagen). Ten embryos per sample were collected in 1.5-mL tubes and lysed in 150 μL of RLT buffer supplemented with 10% of 2-mercaptoethanol with a motorized pestle mixer (Cole-Parmer). RNA was purified according to the manufacturer's instructions. RNA was reverse transcribed to cDNA using the iScript cDNA Synthesis Kit (Bio-Rad). cDNA levels were quantified by RT-qPCR, using the Gotaq qPCR Kit (Promega) and the primers specified in Appendix Table S1.

## RNA-seq

RNA from gonads isolated from wild-type or *eif4e1b* juvenile fish with high expression of *ziwi:GFP* (3 gonads per sample) was prepared using the RNeasy Mini Kit (Qiagen). RNA samples, including RNA from gonads, RIP input and RIP elution samples, were submitted to Macrogen for library preparation and NGS sequencing. RIP input samples (corresponding to total RNA isolated from embryo lysates) and RNA from gonads were prepared with a SMARTer Stranded RNA library (Ribo-Zero) protocol, whereas RIP samples were prepared with SMARTer Ultra Low RNA kit. RNA samples were sequenced using NovaSeq 6000 platform (Illumina). Reads were aligned to the zebrafish genome (GRCz11) and differential expression analysis was performed with the DESeq2 package (Love et al, 2014) in R 3.4.1. Graphs of DEG analyses were generated in RStudio 2021.09.2; genes encoding proteins belonging to the following InterPro families were highlighted in Figs. 5A and 6H: IPR009072 (histone fold), IPR005819 (linker histone H1/H5), IPR001951 (histone H4), IPR019809 (histone H4, conserved site), IPR035425 (CENP-T/histone H4, histone fold), IPR000164 (histone H3/CENP-A), IPR007125 (histone H2A/H2B/H3), IPR000558 (histone H2B), IPR002119 (histone H2A), IPR032454 (histone H2A, C-terminal domain), IPR032458 (histone H2A conserved site), IPR005818 (linker histone H1/H5, domain H15), IPR021171 (core histone macro-H2A), and IPR035796 (core histone macro-H2A, macro domain). Reads were visualized with Integrative Genome Viewer (IGV) version 2.11.9. Comparisons between the data generated in this study with previously published datasets were done in Microsoft Excel version 16.51.

## Tethering assays

3× *MS2* loops were inserted by Gibson assembly before the Xenopus *β-globin* 3' UTR of a vector containing SP6 and T3

promoter sites, the Xenopus *β-globin* 5' UTR, the coding sequence of superfolder GFP (sfGFP), the Xenopus *β-globin* 3' UTR, and an $A^{27}$, $C^{18}$ tail. The *MS2 coat protein* (*MCP*) sequence was introduced without a STOP codon by Gibson assembly before *eIF4Ea* or *eIF4E1b* coding sequences, in vectors containing SP6 and T3 promoters, the Xenopus *β-globin* 5' UTR, a 3xFlag tag, the coding sequence of *eIF4Ea* or *eIF4E1b*, the Xenopus *β-globin* 3' UTR and an $A^{29}$, $C^{14}$ tail.

mRNAs were transcribed in vitro from linearized plasmids using the SP6 mMessanger Machine Kit (Ambion) for *MCP*-containing mRNAs, or the HiScribe SP6 NEB kit and the 3′-O-Me-m⁷G(5′) ppp(5')G RNA Cap Structure Analog (ARCA) for the *GFP* reporter mRNA. All mRNAs were purified using the RNA Clean & Concentrator kit (Zymo Research). 67.5 pg of sfGFP reporter mRNA and an equimolar amount (82.5 pg) of *eIF4Ea* or *eIF4E1b* mRNA were co-injected into one-cell embryos. 67.5 pg of sfGFP reporter mRNA was also injected alone. Ten embryos per condition were collected for RNA extraction and RT-qPCR (see above; Appendix Table S1). Twenty embryos were manually dechorionated and lysed in 100 mM Tris-HCL pH 7.5, 150 mM NaCl, 0.05% Triton X-100 (Sigma-Aldrich) and cOmplete, EDTA-free protease inhibitor cocktail (Merck) with a motorized pestle mixer (Cole-Parmer) for western blotting.

### Experimental design and statistical analysis

All reported experiments were replicated at least twice in the laboratory. Zebrafish were grouped according to genotype and age. Male and female adult fish used to obtain embryos were between 3 months and 1-year-old. In vivo samples were randomly assigned to the experiment and treated equally. The sample size was not statistically predetermined. For in vivo experiments, fish and embryos were randomly selected. To test the effect of different mutations on eIF4E1b localization and the tethering of eIF4E proteins to a reporter mRNA, the experiments were performed blindly.

Statistical analyses and graphs were performed using GraphPad Prism 8.0.2. Standard statistical tests were selected to account for sample normality, and post hoc tests were used for multiple comparisons. Tests and *P* values are shown in the graphs and figure legends.

### Reagent availability

All reagents generated in this study will be distributed upon request.

## Data availability

Previously published structures used in this study are available at RCSB Protein Data Bank (RCSB PDB): 5BXV, 5T46, 4UE9. Translational efficiency and polyA tail length data were previously published and accessible at Gene Expression Omnibus (GEO; accession number: GSE52809) and at https://zenodo.org (names: hs25.h5.xz.part01 to 11), respectively. Available RNA-seq data from different stages of zebrafish embryogenesis and adult tissues: GSE147112 (oogenesis and embryogenesis), GSE32898 (embryogenesis), GSE111882 (ovaries), and GSE171906 (adult tissues and

organs). RIP-seq and RNA-seq data generated in this study is available at GEO with the accession numbers GSE233570 (RIP-seq) and GSE241537 (RNA-seq from gonads). Mass spectrometry proteomics data has been deposited to the ProteomeXchange Consortium via the PRIDE (Perez-Riverol et al, 2019) partner repository with the dataset identifier PXD042434.

## Peer review information

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

## Acknowledgements

We thank A Schleiffer (Bioinformatics, IMP) for performing the phylogenetic analysis of eIF4E genes; M Novatchkova (Bioinformatics, IMP) and J. Elbers for their help with RIP-seq and RNA-seq data analysis; P Bhat for analyzing the TAIL-seq data; C Igreja (MPI for Biology, Tübingen) for providing the plasmids for the expression of the human eIF4E-binding motifs of eIF4G, eIF4EBP1 and 4E-T in *E. coli*; B Draper (UC Davis) for providing the *ziwi:EGFP* plasmid for transgenesis; the Proteomics facility at the Vienna BioCenter Core Facilities (VBCF), in particular O Hudecz, R Imre, S Opravil, K Mechtler and E Roitinger, for the processing, analyses and supervision of MS data; the Histology facility at VBCF, in particular L Ushakova and M Grivej, for the staining of zebrafish ovaries; the BioOptics facility at IMP, in particular T Lendl, P Pasierbek and A Moreno-Cencerrado, for their help with microscopy data acquisition and analysis; the IMP animal facility, especially F Puhl, K Rattner, J König, and D Sunjic for their care of zebrafish; T Mylenko, A Bandura, G Deneke, and M Binner for their help with genotyping; the Molecular Biology Service at IMP for Sanger sequencing and for providing competent cells and reagents; J Ahel for implementing AF Multimer in the VBC cluster; the entire Pauli lab, M Madeira Reimao Pinto, and the VBC RNA Salon for critical feedback and valuable discussions; U Hohmann, A Chugunova, L Veleti, and C Plaschka for feedback on the manuscript. Research in the lab of A. Pauli was funded by the Research Institute of Molecular Pathology (IMP), Boehringer Ingelheim, the Austrian Academy of Sciences, FFG (Headquarter grant FFG-852936), the FWF START program (Y 1031-B28), the ERC Consolidator grant 101044495/GaMe, an HFSP Career Development Award (CDA00066/2015), an HFSP Young Investigator Grant (RGY0079/2020) and the FWF SFB RNADeco (project number F80). L Lorenzo-Orts was supported by an SNF Early Postdoc.Mobility fellowship (P2GEP3_191204), an EMBO long-term fellowship (ALTF 1165-2019), and an MSCA-IF-EF-SE (890218).

## Author contributions

**Laura Lorenzo-Orts**: Conceptualization; Data curation; Formal analysis; Supervision; Funding acquisition; Validation; Investigation; Visualization; Methodology; Writing—original draft; Project administration; Writing—review and editing. **Marcus Strobl**: Data curation; Validation; Investigation; Visualization; Methodology. **Benjamin Steinmetz**: Data curation; Formal analysis; Investigation; Methodology. **Friederike Leesch**: Methodology. **Carina Pribitzer**: Methodology. **Josef Roehsner**: Methodology. **Michael Schutzbier**: Resources; Methodology. **Gerhard Dürnberger**: Resources; Data curation; Formal analysis; Supervision. **Andrea Pauli**: Conceptualization; Resources; Data curation; Supervision; Funding acquisition; Writing—original draft; Project administration; Writing—review and editing.

## Disclosure and competing interests statement

The authors declare no competing interests.

# Expanded View Figures

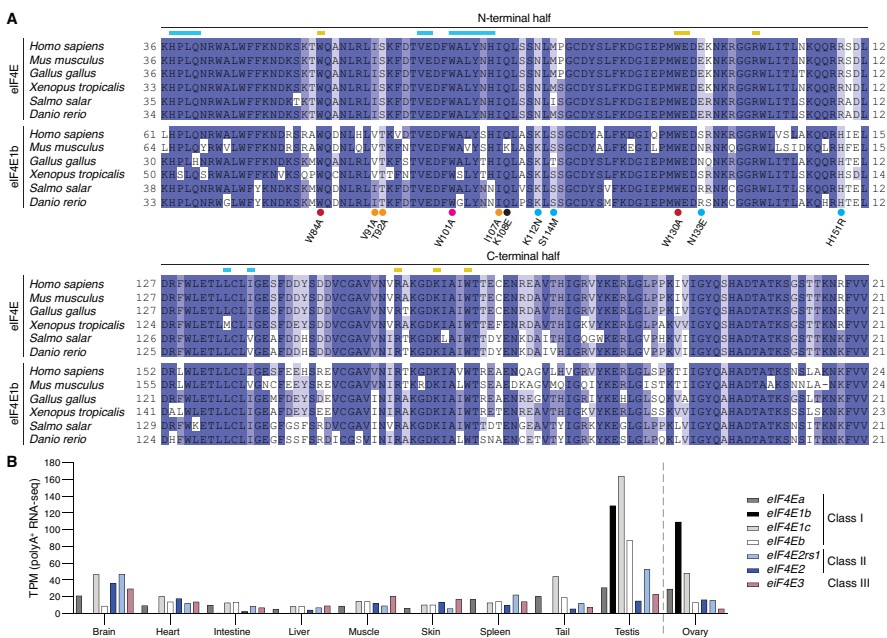

**Figure EV1. eIF4E1b and eIF4E are highly conserved but have specific expression patterns.**

(**A**) Amino acid alignment of eIF4E and eIF4E1b proteins from six vertebrate species. Unstructured N-terminal regions (see Fig. 1A), which are not included in the constructs used for expressing recombinant proteins, are excluded from the alignment. Residues interacting with the mRNA cap or with eIF4E-binding motifs are highlighted with yellow or blue lines, respectively. eIF4E1B residues mutated in Fig. 3D are indicated with dots (red: mRNA cap binding; pink: dorsal site; orange: lateral site; blue: conserved in eIF4E but different in eIF4E1b proteins; black: others). N-terminal and C-terminal regions exchanged in chimeric constructs are indicated. (**B**) mRNA levels (in transcripts per million, TPM) of zebrafish eIF4Es in different organs and adult tissues based on polyadenine-selected RNA-seq data (Fujihara et al, 2021) (ovary RNA-seq data from Herberg et al, 2018).

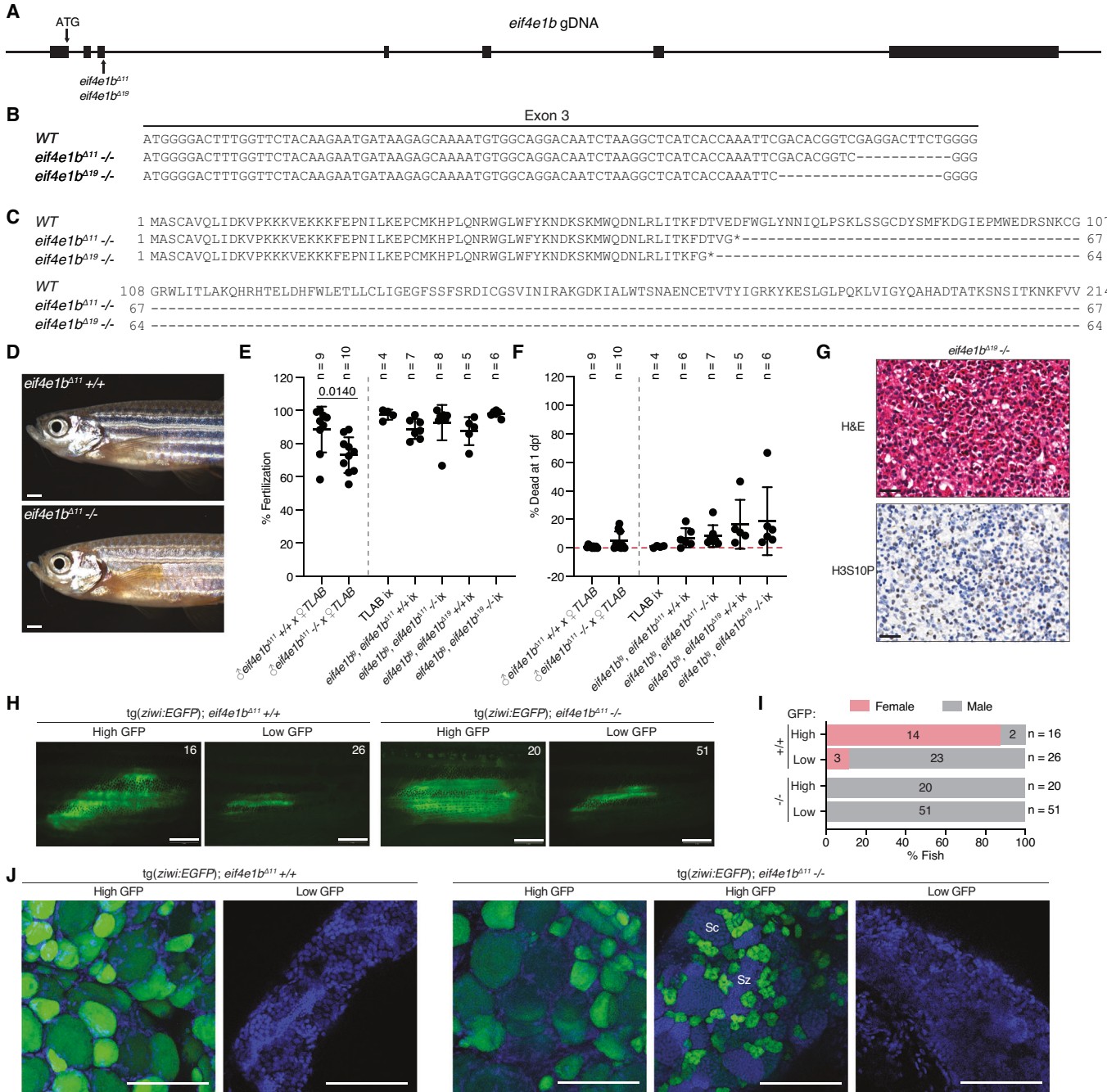

**Figure EV2. *eif4e1b* mutants develop into fertile males due to sex reversal.**

(**A**) Scheme of the zebrafish *eif4e1b* locus, indicating the start codon (*ATG*) and the 11 and 19 nt deletions present in the two *eif4e1b* mutants (*eif4e1b^Δ11^* and *eif4e1b^Δ19^*, respectively) generated in this study. Exons are depicted as boxes, introns as lines. (**B**) Nucleotides missing in the third exon of *eif4e1b* mutants. (**C**) eIF4E1b mRNA translation in *eif4e1b* mutants result in truncated proteins of 67 (for *eif4e1b^Δ11^*) and 64 (for *eif4e1b^Δ19^*) amino acids. Asterisks indicate premature stop codons. (**D**) Representative images of wild-type and *eif4e1b^Δ11^* mutant sibling males obtained from a heterozygous incross (scale bars = 1 mm). (**E**) Fertilization rates of embryos obtained by crossing *eif4e1b^Δ11^* homozygous or wild-type males with wild-type females (left). Overexpression of *3xflag-sfGFP-eIF4E1b* (*eif4e1b^tg^*) in transgenic wild-type and *eif4e1b* mutant siblings results in fertile males and females (right; ix = incross). (**F**) Percentage of dead embryos at 1-day post fertilization (dpf). Embryos were obtained by crossing homozygous or wild-type *eif4e1b^Δ11^* male siblings with wild-type females (left), or by incrossing (ix) wild-type or mutant fish expressing *3xflag-sfGFP-eIF4E1b* (right). (**G**) Hematoxylin and eosin (H&E, top) and phospho-histone H3 (bottom) staining of *eif4e1b^Δ19^* ovary sections. Scale bars = 20 μm. (**H**) Live microscopy of gonads from juvenile wild-type and *eif4e1b^Δ11^* mutant siblings expressing eGFP under the control of the *ziwi* promoter (scale bars = 1 mm). Fish were classified based on the area and intensity of eGFP. Numbers (top right) indicate the number of fish in each category. (**I**) Number of fish in *H* that developed into males or females. (**J**) Confocal microscopy of fixed gonads isolated from wild-type (left) and *eif4e1b^Δ11^* homozygous (right) juveniles expressing eGFP under the control of the *ziwi* promoter. Nuclei were stained with DAPI (in blue; Sc = spermatocytes; Sz = spermatozoa; scale bars = 100 μm). Data information: (**E, F**) lines indicate mean with SD. Significance for the first two genotypes was calculated using unpaired *t* tests. The other genotypes were compared using one-way ANOVA followed by Tukey's test (if not indicated, *P* value > 0.005).

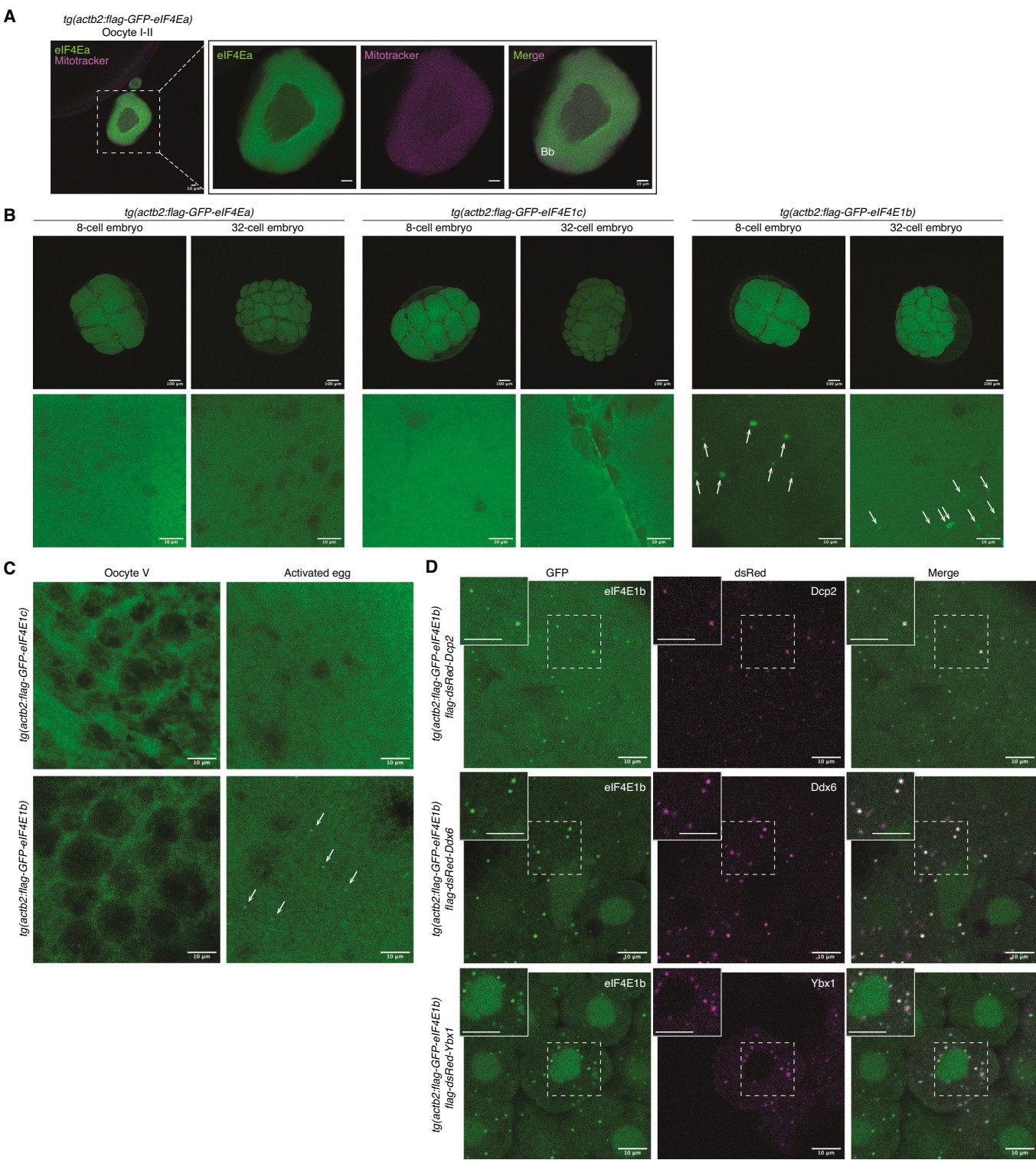

**Figure EV3. eIF4E1b localizes to P-bodies in the embryo.**

(A) eIF4Ea localizes to the cytosol of fixed early zebrafish oocytes. Mitochondria present in the Balbiani body (Bb) are stained with Mitotracker. Scale bars = 10 μm. (B) Confocal microscopy images of fixed transgenic embryos expressing 3xflag-sfGFP-tagged eIF4Ea, eIF4E1b and eIF4E1c (scale bars: top = 100 μm; bottom = 10 μm). eIF4E1b foci are indicated by white arrows. (C) Confocal microscopy images of squeezed (oocyte V) and activated eggs from transgenic lines expressing 3xflag-sfGFP-tagged eIF4E1c (top) and eIF4E1b (bottom). eIF4E1b foci are indicated by white arrows. (D) Colocalization experiments in zebrafish embryos expressing *3xflag-sfGFP-eIF4E1b* transiently expressing P-body markers. mRNAs containing the coding sequences of dsRed-tagged P-body components (Dcp2, Ddx6 and Ybx1) were injected into 1-cell embryos. Images were taken at 3 h post fertilization. Regions enclosed by dashed boxes are shown at higher magnification on the top left (scale bars = 10 μm).

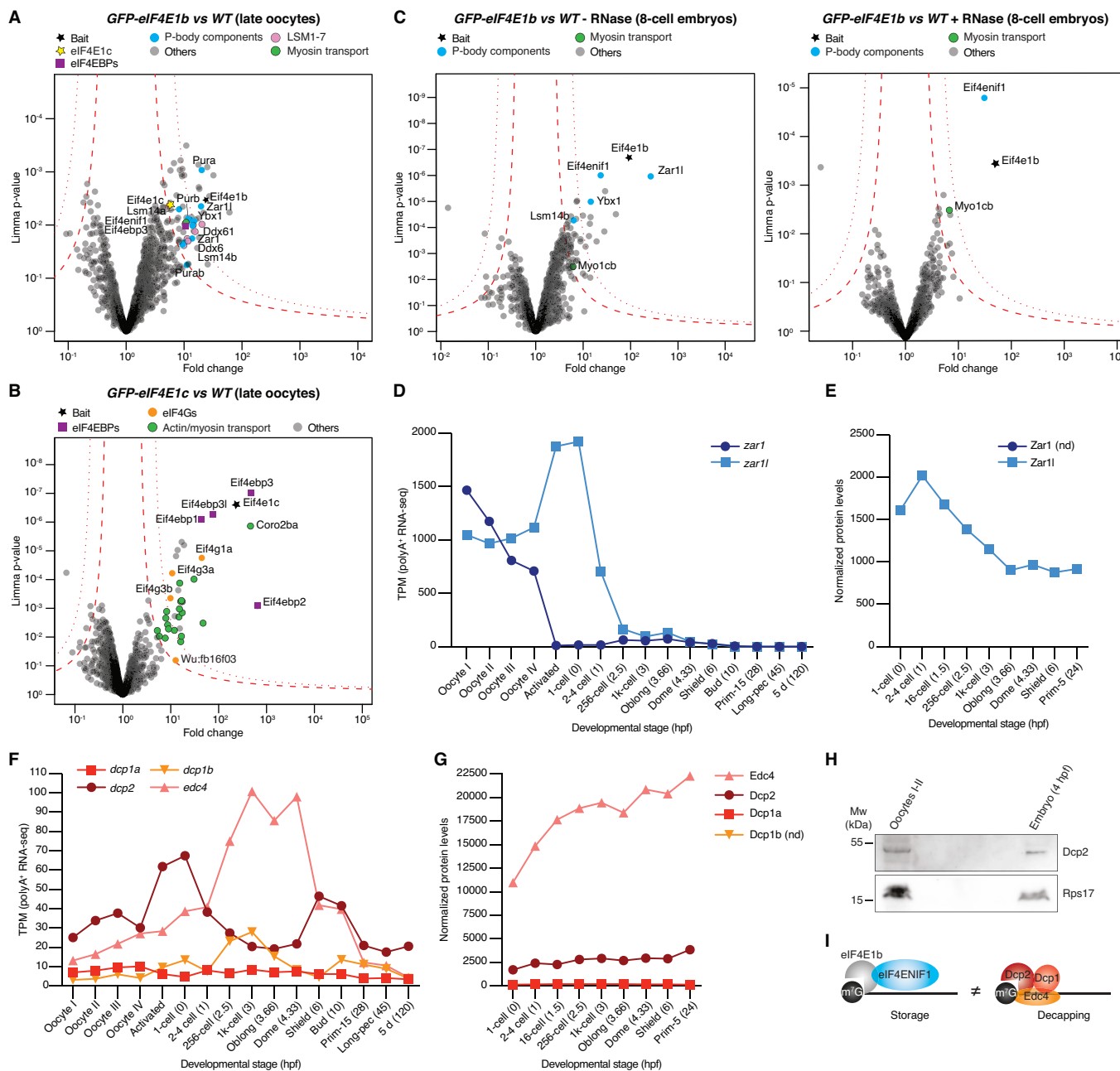

**Figure EV4. Additional IP-MS and expression data.**

(A, B) Volcano plots of the proteins identified by IP-MS in zebrafish late oocytes (stage III–IV) expressing GFP-tagged eIF4E1b (A) or eIF4E1c (B) compared to wild type (WT). (C) Volcano plots of the proteins identified by IP-MS in 8-cell embryos expressing GFP-tagged eIF4E1b compared to WT in the absence (mock, left) or presence (right) of RNase I. (D) Levels of zar1 and zar1l mRNAs during zebrafish oogenesis and embryogenesis based on polyA-selected RNA-seq data (Pauli et al, 2012; Cabrera-Quio et al, 2021). (E) Normalized expression of Zar1l protein during zebrafish embryogenesis in TMT-MS. Zar1 peptides were not detected (nd). (F, G) mRNA (F) and protein (G) levels of decapping factors during zebrafish oogenesis and embryogenesis based on polyA-selected RNA-seq (Pauli et al, 2012; Cabrera-Quio et al, 2021) and TMT-MS data, respectively. (H) Western blot showing the expression of Dcp2 (predicted Mw of 45.5 kDa) and Rps17 (loading control, predicted Mw of 15.4 kDa) in zebrafish early oocytes and embryos. Data information: (A–C) statistical significance was determined using limma (Smyth, 2005). Permutation-based false discovery rates (FDRs) are displayed as dotted (FDR < 0.01) or dashed (FDR < 0.05) lines (n = 3 biological replicates). Hpf hours post fertilization, Mw molecular weight, TPM transcripts per million. (I) Model of eIF4E1b function in mRNA stability: eIF4E1b interacts with eIF4ENIF1 in P-bodies and binds to the mRNA cap (left), thereby blocking access to the decapping machinery (right), which is also located in P-bodies.

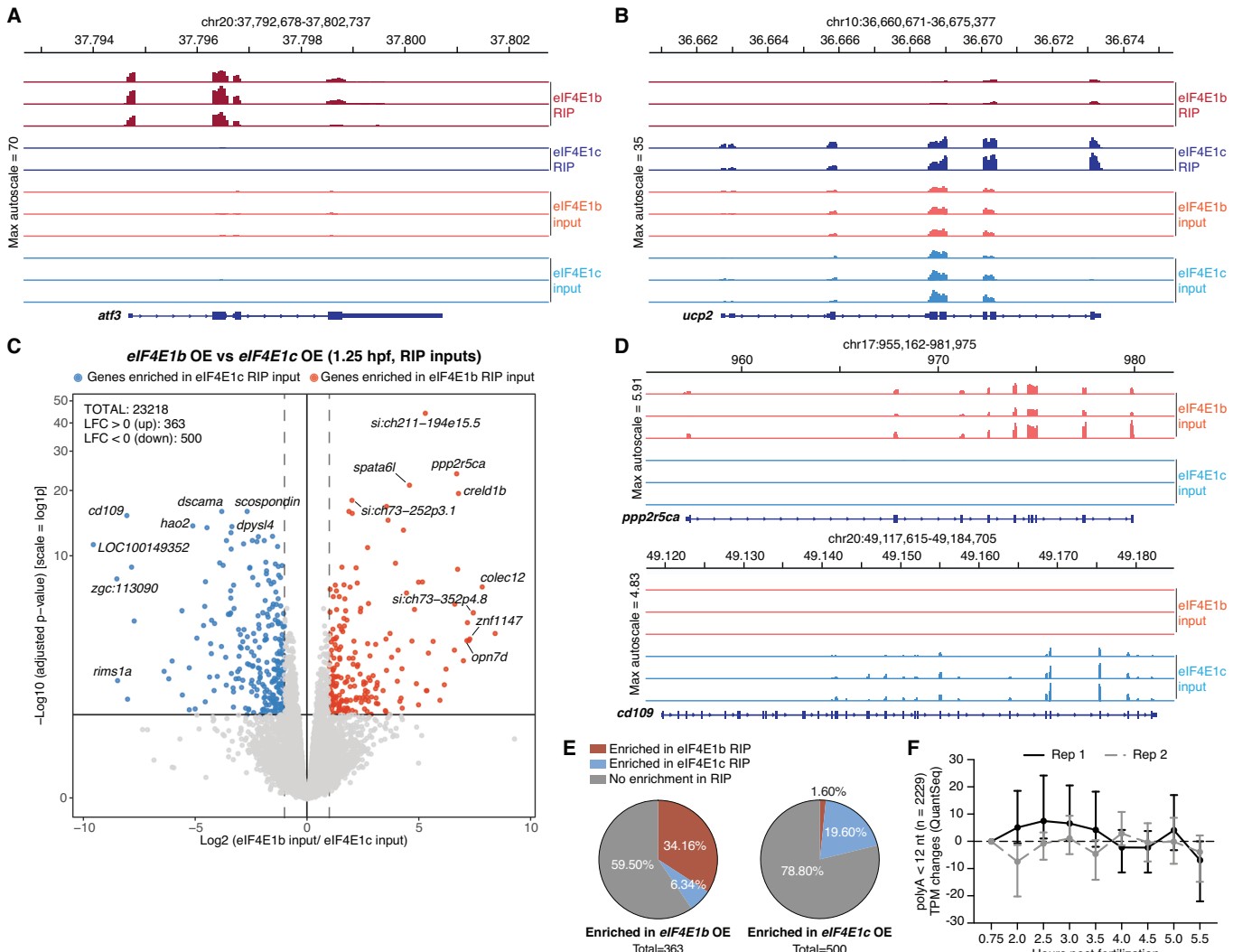

**Figure EV5. Additional RNA immunoprecipitation (RIP) and RNA sequencing (RNA-seq) analyses.**

(A, B) Examples of mapped RNA-seq reads of differentially expressed genes (DEGs) in eIF4E1b (A) and eIF4E1c (B) RIPs performed at 1.25 hours post fertilization (hpf). (C) DEG analyses of mRNAs isolated from total lysates (inputs from RIP experiments) of 8-cell stage embryos expressing 3xflag-sfGFP tagged eIF4E1b or eIF4E1c under the control of the *actb2* promoter (*eIF4E1b* or *eIF4E1c* OE, respectively). mRNAs enriched in *eIF4E1b* OE and *eIF4E1c* OE are shown in red and blue, respectively. Statistical significance was determined using Benjamini-Hochberg-corrected Wald test (DeSeq2; *P* value < 0.005; *n* = 3 biological replicates). (D) Mapped RNA-seq reads of example genes specifically upregulated in *eIF4E1b* (top) or *eIF4E1c* (bottom) OE 8-cell embryos. (E) Fraction of transcripts significantly upregulated in *eIF4E1b* OE (left) or *eIF4E1c* OE (right) embryos that were also enriched in eIF4E1b and eIF4E1c RIPs at 1.25 hpf (*P* value < 0.05). (F) Changes in mRNA levels (in transcripts per million, TPM) of transcripts with polyA tails containing less than 12 nucleotides during zebrafish embryogenesis according to published RNA-seq data (Bhat et al, 2023). Medians (dots) with interquartile ranges (bars) are shown.

