## [Peer Review File · EMBO Reports]

eIF4E1b is a non-canonical eIF4E protecting maternal dormant mRNAs

Laura Lorenzo-Orts, Marcus Strobl, Benjamin Steinmetz, Friederike Leesch, Carina Pribitzer, Josef Roehsner, Michael Schutzbier, Gerhard Dürnberger, and Andrea Pauli

DOI: [10.15252/embr.202357610](https://doi.org/10.15252/embr.202357610)

Corresponding author(s): *Andrea Pauli (andrea.pauli@imp.ac.at)* , *Laura Lorenzo-Orts (laura.lorenzo@imp.ac.at)*

Review Timeline:

Submission Date:	7th Jun 23
Editorial Decision:	29th Jun 23
Revision Received:	28th Sep 23
Editorial Decision:	24th Oct 23
Revision Received:	31st Oct 23
Accepted:	8th Nov 23

Editor: *Esther Schnapp*

Transaction Report:

Dear Andrea,

Thank you for the submission of your manuscript to EMBO reports. We have now received the full set of referee reports that is pasted below.

As you will see, the referees acknowledge that the findings are potentially interesting. However, they also all point out that the data are not sufficiently strong to support the main conclusions. All referees pinpoint that it remains unclear what happens to eIF4E1B-bound mRNAs in the absence of eIF4E1B and/or later in development. I think that stronger data will be required to proceed with the manuscript here. All referee comments seem reasonable and I would like to ask you to address all. Please let me know in case you disagree, and we can discuss the revisions further, also in a video chat, if you like.

I would thus like to invite you to revise your manuscript with the understanding that the referee concerns must be fully addressed and their suggestions taken on board. Please address all referee concerns in a complete point-by-point response. Acceptance of the manuscript will depend on a positive outcome of a second round of review. It is EMBO reports policy to allow a single round of major revision only and acceptance or rejection of the manuscript will therefore depend on the completeness of your responses included in the next, final version of the manuscript.

We realize that it is difficult to revise to a specific deadline. In the interest of protecting the conceptual advance provided by the work, we recommend a revision within 3 months (29th Sep 2023). Please discuss the revision progress ahead of this time with the editor if you require more time to complete the revisions.

- 1) A data availability section providing access to data deposited in public databases is missing. If you have not deposited any data, please add a sentence to the data availability section that explains that.
- 2) Your manuscript contains statistics and error bars based on $n=2$. Please use scatter blots in these cases. No statistics should be calculated if $n=2$.

5) a complete author checklist, which you can download from our author guidelines

<<https://www.embopress.org/page/journal/14693178/authorguide>>. Please insert information in the checklist that is also reflected in the manuscript. The completed author checklist will also be part of the RPF.

6) Please note that all corresponding authors are required to supply an ORCID ID for their name upon submission of a revised manuscript (<<https://orcid.org/>>). Please find instructions on how to link your ORCID ID to your account in our manuscript tracking system in our Author guidelines <<https://www.embopress.org/page/journal/14693178/authorguide#authorshipguidelines>>

7) Before submitting your revision, primary datasets produced in this study need to be deposited in an appropriate public database (see <https://www.embopress.org/page/journal/14693178/authorguide#datadeposition>). Please remember to provide a reviewer password if the datasets are not yet public. The accession numbers and database should be listed in a formal "Data Availability" section placed after Materials & Method (see also <https://www.embopress.org/page/journal/14693178/authorguide#datadeposition>). Please note that the Data Availability Section is restricted to new primary data that are part of this study. * Note - All links should resolve to a page where the data can be accessed. *
If your study has not produced novel datasets, please mention this fact in the Data Availability Section.

I look forward to seeing a revised form of your manuscript when it is ready.

Best wishes,
Esther

Referee #1:

This study aims to characterize the cap binding protein eIF4E1b combining experiments with semi-purified proteins in vitro and experiments in zebrafish extracts. Key findings include:

- Mouse eIF4E1b binds the mRNA cap and interacts preferentially with the translational repressor eIF4ENIF1/4E-T rather than the translation initiation factor eIF4G in vitro.
- Zebrafish eIF4E1b is abundant in the cytoplasm and in puncta which also contain P-body component DDX6. P body localization depends on residues required for interaction with the translational repressor and P body component eIF4ENIF1/4E-T.
- Zebrafish eIF4E1b immunoprecipitates with P body components and with mRNAs that on average have short polyA tails early in development and are stable during the oocyte to embryo transition (based on analyses performed in prior studies).
- A knock out of eIF4E1b causes sex reversal from females to sterile males, suggestive of a function early in female germline development.

Based on these findings, the authors propose that eIF4E1b is required to store dormant, translationally repressed maternal transcripts required for embryonic development. These findings are consistent with a recent study focused on mouse eIF4E1b (Lee et al., 2023), but unfortunately are not supported by the data presented. Most importantly there are no functional assays that directly demonstrate a requirement for eIF4E1b for mRNA stability or translational repression.

I suggest the authors modify the title of the paper, abstract and discussion to avoid speculative statements such as "Here we report an essential role for the germline-specific paralog of the mRNA cap-binding factor eIF4E, known as eIF4E1b, in the storage and repression of maternal mRNAs with short polyA tails", since the data are only correlative.

The authors should also include in their introduction prior work in *Drosophila* and *C. elegans* implicating eIF4Es in mRNA storage/translational repression in oocytes (for example <https://pubmed.ncbi.nlm.nih.gov/25179781/>, <https://pubmed.ncbi.nlm.nih.gov/28701521/>, <https://pubmed.ncbi.nlm.nih.gov/32079657/>)

Figure 2 and Figure 3:

- the species of origin for each protein should be indicated. The authors should also explain in greater detail their assays and how they identified each band of the gels shown.
- What is the difference/significance between the oocyte/embryo pictures shown in Fig. 2E and F and Fig. 3G???
- Fig. 3H reports on the relative number of granules relative to wild-type - it would be more helpful to report on the actual number of granules to evaluate the data directly.
- Fig. 3H reports that the cap binding mutant eIF4E1b localizes to granules as efficiently as WT. The authors may want to point out this finding which suggests that the eIF4E1b/P body interaction does not require cap binding/binding to RNA?

Fig. 5D and E: the polyA tail length of mRNAs bound by eIF4E1b are shorter on average during early development and become longer with time, eventually reaching average sizes. Because the authors only perform RIP at one stage (8-cell - early development), whether eIF4E1b continues to bind the mRNAs when they become polyadenylated and translated is not known. Absent more information (direct loss of function analyses?), any conclusion as to the function of eIF4E1b appears premature. In fact the finding that eIF4E1b is required early in female oogenesis suggests an essential function BEFORE the oocyte to embryo transition.

Referee #2:

In this manuscript, Lorenzo-Orts et al. aim to uncover how maternal mRNAs, stored during oogenesis with short polyA tails, avoid degradation.

The scientific premise pointing to eIF4Es as protein candidates to be in charge of the maternal mRNA dormancy is speculative, but the authors zero in the eIF4E1b due to its high expression during oogenesis and in the early embryo. Using an elegant CRISPR/Cas9 strategy, the authors show that upon loss of eIF4E1b, there is impaired oocyte development

and sex reversal - no females are formed. The few female-looking fish have no oocytes but a homogeneous cell mass. The authors demonstrate that eIF4E1b from mouse interacts with the cap structure of mRNAs and with the proteins eIF4ENIF1 and eIF4EBP1, but no eIF4G, excluding a role for eIF4E1b in promoting translation. This data contradicts previous publications on eIF4E1b but the confidence in these results is based on the fact that eIF4E1b contains the key aminoacids that mediate these interactions.

Next the authors conduct immunoprecipitation of eIF4E1b from transgenic embryos and identify the partners by mass spectrometry. This data reveals that eIF4E1b interacts with many p-body components but not the decapping enzymes, suggesting that eIF4E1b target maternal mRNAs to p-bodies yet prevents their decapping and degradation.

To identify the targets of eIF4E1b, the authors conduct a RIP-Seq experiment. It reveals that the mRNA targets of eIF4E1b, are stabilized by the interaction and that are transnationally repressed, despite they show elongation of the polyA tail.

Overall, this is a rigorous work showcasing eIF4E1b as a novel factor involved in the translation repression and stability of maternal mRNAs. The data is well presented and the experimental strategy is rigorous, which includes establishing several zebrafish transgenic and mutant lines. The results from this work will be the seed for future studies aiming at understanding long-term stability of maternal mRNAs. Therefore, the manuscript is adequate for publication in EMBO Reports if the authors address the following question:

The overarching goal of the manuscript is to uncover the factor responsible for the dormancy of maternal mRNA. While indeed the authors show how eIF4E1b binds to capped mRNAs and prevents their translation by localization to P-bodies and their decay by not interacting with decapping enzymes, it is not clear if this regulation mediated by eIF4E1b is dynamic and developmentally regulated. It would be expected that the factor involved in silencing maternal mRNAs would relieve its repression at some point during early embryogenesis (e.g. onset of zygotic transcription, maternal-to-zygotic transition) to ensure normal development. The authors base their analysis in samples during late oogenesis or at 8-cell stage embryos. Figure 1D shows how the eIF4E1b protein levels decay by shield stage. The authors should determine if eIF4E1b loses affinity for mRNA at later stages of embryogenesis, and if the mRNAs associated with eIF4E1b eventually recover translation potential.

Referee #3:

Lorenzo-Ortiz, L et al. in this manuscript discuss a role for eIF4E1B in the stability and translation repression of maternally deposited mRNAs in zebrafish. First, the authors generated KO zebrafish lines and showed that eIF4E1B also plays a role in ovary development, leading to male bias in homozygous KO fish (Fig 1). Then, the authors tested several eIFs and their capability to bind to mRNA's cap and other eIFs (Fig 2). Since eIF4E1B interacts with eIF4ENIF1, known to localize to P-bodies, the authors show through microscopy puncta expression of eIF4E1B but not of other eIFs (Fig 2). To explore such protein interactions, the authors performed several binding predictions, that were later validated in vitro and in vivo with mutant versions of eIF4E binding motifs, showing loss of puncta expression in injected zebrafish embryos (Fig 3). Further, authors looked for the protein interactome of eIF4E1B using IP followed by mass spectrometry and found enrichment in P-body components, but not of translation factors (Fig 4). Finally, authors employed RNA immunoprecipitation and identified that eIF4E1B binds to maternal mRNAs with short polyA tail hinting its function as prevention of decay and translation (based on RNA-seq and ribosome profiling datasets) (Fig 5).

Major:

Overall, the manuscript is well written, and figures correctly address questions posed by the authors. In addition, authors elegantly show eIF4E1B protein interaction with other eIFs and generated IP datasets will be valuable resources for others interested in studying translation regulation in early zebrafish embryogenesis. Nevertheless, it was recently shown in mice that deletion of eIF4E1B causes arrest at the 2-cell stage (Yang, G. et al. 2023 - Genes & Development; Guo, J. et al., 2023 - Advanced Science) (both cited in the manuscript), yet the authors do not provide results or discussion if a similar arrest in development happens in the generated zebrafish knock-out lines. If eIF4E1B is instrumental for maternal mRNA dormancy and protection in zebrafish, then I would expect that the loss of function mutants show early developmental problems either before or at the onset of genome activation. Moreover, authors imply a major role of eIF4E1B in the stability of maternal mRNAs with short poly-A tail, yet there is no data showing that the absence of eIF4E1B causes decay of these mRNAs. Most of the data raise from RIP-seq experiments follow by the characterization of targets. In other words, eIF4E1B might interact with genes depending in the poly(A)-tail length but might not have any role during early development. To show that those correlation are actually part of the eIF4E1B, it would be nice to measure poly(A)-tail length, translation (Ribosome profiling) and mRNA decay profiles in the mutant and wild-type embryos. Indeed, if such an effect is not achieved in zebrafish it might be because eIF4E1B is not as instrumental in zebrafish as it is in mice or it might be redundancy problem. Could it be that in mice the protein is not as highly deposited as in zebrafish?

While it is very attractive to propose the eIF4E1B play a role in maternal mRNA dormancy and be part "fundamental post-transcriptional regulatory principles governing early vertebrate Development" (abstract). Based on the results (phenotypes), it looks that eIF4E1B play a fundamental role in female germline development in zebrafish. Maybe this should be the title of the work.

Minor:

1) In Figure 1 the authors claim that eIF4E1B is highly expressed in zebrafish oogenesis and early embryogenesis, but panels C

and D only show expression levels of selected eIF4Es in comparison to each other. How highly expressed is eIF4E1B when compared to the whole transcriptome and proteome? Can the authors add a reference line of for example the median of the total transcriptome and proteome for a better comparison?

2) There is a typo in line number 360: "were shacked at 800 rpm", which should be "shaken" (verb for motion).

3) Supplemental excel tables lack description about their contents and how they are organized (meaning of colors, what the summary sheet stands for, etc).

4) As a dataset resource, it would be ideal to add o with all identified proteins from mass spectrometry proteomics data not filtered by p-value of FC analysis performed in the tables.

Response to reviewer comments:

We thank the reviewers for their time and effort in reviewing our manuscript. Below we provide point-by-point responses. Comments from reviewers are in **blue**, while comments from us (authors) are in **black**.

Referee #1:

This study aims to characterize the cap binding protein eIF4E1b combining experiments with semi-purified proteins in vitro and experiments in zebrafish extracts. Key findings include:

- Mouse eIF4E1b binds the mRNA cap and interacts preferentially with the translational repressor eIF4ENIF1/4E-T rather than the translation initiation factor eIF4G in vitro.
- Zebrafish eIF4E1b is abundant in the cytoplasm and in puncta which also contain P-body component DDX6. P body localization depends on residues required for interaction with the translational repressor and P body component eIF4ENIF1/4E-T.
- Zebrafish eIF4E1b immunoprecipitates with P body components and with mRNAs that on average have short polyA tails early in development and are stable during the oocyte to embryo transition (based on analyses performed in prior studies).
- A knock out of eIF4E1b causes sex reversal from females to sterile males, suggestive of a function early in female germline development.

Based on these findings, the authors propose that eIF4E1b is required to store dormant, translationally repressed maternal transcripts required for embryonic development. These findings are **consistent with a recent study focused on mouse eIF4E1b** (Lee et al., 2023), but unfortunately are not supported by the data presented. Most importantly there are **no functional assays** that directly demonstrate a requirement for eIF4E1b for mRNA stability or translational repression.

We thank the reviewer for summarizing our findings. As pointed out by the reviewer, our overall finding that eIF4E1b is important during the egg-to-embryo transition is consistent with prior findings in mice. However, our conclusions about the function of eIF4E1Bs are in stark contrast to two studies published this year in mice (Guo et al. 2023 and Yang et al. 2023), which propose that eIF4E1B promotes the translation of specific transcripts during the mouse oocyte-to-embryo transition. Guo et al. suggest a model in which eIF4E1B binds to eIF4G and triggers eIF4F-dependent translation: *“These results suggest that it is possible to assume a potential oocyte-specific closed-loop-model: eIF4E1B-eIF4G-PABPC1L, which can be used to control various target genes to promote the stability and translation efficiency of multiple developmental events during OET”*. Nevertheless, the proposed interaction between eIF4E1B and eIF4G is not supported by experimental evidence in any of the two papers. In fact, mouse eIF4E1B does not coimmunoprecipitate with eIF4Gs but with P-body components such as Zar1 (Guo et al. 2023) and eIF4ENIF1 (Yang et al. 2023), in line with our in vivo (IP-MS in zebrafish) and in vitro (pulldowns) data with zebrafish and mammalian eIF4E1B, respectively. Several studies have shown that P-bodies contain translationally repressed mRNAs (Bregues et al. 2005; Halstead et al. 2015; Hubstenberger et al. 2017). Moreover, both Zar1 and eIF4ENIF1 have been proposed to act as translational repressors (Yamamoto et al. 2013; Miao et al. 2017; Rong et al. 2019; Kamenska et al. 2014; Räscher et al. 2020), thereby challenging the idea of eIF4E1B functioning in translation initiation, as Guo et al. and Yang et al. propose.

We fully agree with the reviewer that obtaining direct evidence for the ability of eIF4E1b to repress translation and stabilize mRNAs would strengthen our conclusions. We had not previously performed these experiments due to practical limitations in obtaining zebrafish eggs depleted of maternal eIF4E1b. We have now performed a series of additional experiments that circumvent this challenge and strengthen our conclusions:

1) **Proteomics of *eIF4E1b* and *eIF4E1c* overexpressing embryos (Fig 6A-F):** *eIF4E1c* overexpression leads to a major enrichment of proteins in the embryo. Using some of the upregulated proteins as examples, we show by RT-qPCR that the upregulation is not at the mRNA level, consistent with eIF4E1c promoting translation. In contrast, overexpression of *eIF4E1b* leads to the dysregulation of 792 proteins, 76% of which are depleted. The levels of mRNAs encoding some of these proteins were similar in both wild-type and *eIF4E1b* overexpressing embryos, in line with eIF4E1b repressing translation. Moreover, according to published data, mRNAs encoding proteins depleted in *eIF4E1b* overexpressing embryos are efficiently translated in wild type. Overall, these new results support our conclusion that eIF4E1b does not promote translation, but rather represses it.

2) **Proteomics and transcriptomics of *eif4e1b* mutant gonads (Fig 6G-I and Appendix Figure S4).** Since loss of eIF4E1b results in sex reversal in zebrafish, we used our *ziwi:GFP* transgenic line to isolate female gonads from *eif4e1b* mutant and wild-type juveniles before sex reversal occurs in *eif4e1b* mutants. It should be noted that we can only obtain *eif4e1b* homozygous mutants by crossing heterozygous fish. Thus, 25% of the fish were of the desired genotype and only about half of them contained female gonads based on high *ziwi:GFP* expression. Proteomic and transcriptomic analyses of these gonads revealed a large fraction of proteins upregulated in *eif4e1b* mutant gonads (~85% of the dysregulated proteins), the majority of which were not upregulated at the mRNA level. Furthermore, many of the mRNAs depleted in *eif4e1b* mutant gonads were enriched in the eIF4E1b RIP, such as histone mRNAs, in line with eIF4E1b stabilizing target mRNAs. Taken together, these new results support the idea that eIF4E1b represses translation and stabilizes mRNAs in vivo.

3) **Binding of eIF4E1b or eIF4Ea to a *GFP* reporter mRNA (Fig 5F-I).** To study the direct impact of eIF4Es on translation, we used the MS2/MCP system to tether canonical eIF4Ea or eIF4E1b to a *GFP* reporter mRNA in the zebrafish embryo. We artificially tethered these proteins to the 3' UTR of a reporter mRNA because (1) MS2 loops in the 5' UTR blocked translation in our hands, and (2) eIF4G binding to the 3' UTR has been reported to promote translation of the upstream open reading frame (Paek et al 2015). While canonical eIF4Ea promoted translation of the reporter mRNA, eIF4E1b did not.

The new experiments described above, as well as our previous data showing that eIF4E1b does not interact with eIF4Gs but with known translation repressors, localizes to P-bodies in the embryo, and binds to mRNAs with reported median polyA tail lengths and translation efficiencies below average, support the idea that eIF4E1b does not promote translation. We believe that clarifying previous claims that have highlighted eIF4E1B as a translational activator without providing direct evidence for it will be of great importance to the scientific community.

I suggest the authors modify the title of the paper, abstract and discussion to avoid speculative statements such as "Here we report an essential role for the germline-specific paralog of the mRNA

cap-binding factor eIF4E, known as eIF4E1b, in the storage and repression of maternal mRNAs with short polyA tails", since the **data are only correlative**.

We thank the reviewer for this comment. We have modified the abstract as suggested and it now reads:

"Here we report an essential role for the germline-specific paralog of the mRNA cap-binding factor eIF4E, known as eIF4E1b, in the storage and repression of maternal mRNAs. eIF4E1b binds to the mRNA cap and is targeted to P-bodies through its direct interaction with eIF4ENIF1/4E-T. In early embryos, eIF4E1b binds to a specific set of mRNAs that are reported to be poorly translated and to contain short or no polyA tails."

The authors should also include in their introduction prior work in *Drosophila* and *C. elegans* implicating eIF4Es in mRNA storage/translational repression in oocytes (for example <https://pubmed.ncbi.nlm.nih.gov/25179781/>, <https://pubmed.ncbi.nlm.nih.gov/28701521/>, <https://pubmed.ncbi.nlm.nih.gov/32079657/>).

We thank the reviewer for the suggestion. We have now included citations to previous work on invertebrates in the *Introduction* and *Discussion* sections:

"In invertebrates, certain eIF4Es play important roles in germline development, such as IFE-1 in *C. elegans* (Henderson et al, 2009; Huggins et al, 2020) and eIF4E-5 in *Drosophila* (Shao et al, 2023), both of which are required for spermatogenesis."

"In *Drosophila*, the eIF4E-binding protein Cup, ortholog to eIF4ENIF1, has also been reported to target maternal *oskar* mRNA to P-bodies (Bayer et al, 2023) and to be essential for fly oogenesis (Lehmann & Nüsslein-Volhard, 1986; Nakamura et al, 2004). We propose a similar mechanism in vertebrates, where eIF4ENIF1-mediated repression of eIF4E1B-bound maternal mRNAs in P-bodies is important for oogenesis."

Figure 2 and Figure 3:

- the species of origin for each protein should be indicated. The authors should also explain in greater detail their assays and how they identified each band of the gels shown.

We have now added the species of origin of the proteins used for in vitro pulldowns in Fig 2 and Fig 3, as well as their predicted molecular weights in the legends of Fig 2, Appendix Figure S2 and Appendix Figure S3. To ensure the identity of each band, we also show lysate samples for each of the proteins, as well as pulldown samples containing only the bait (an eIF4E) or the prey (an eIF4E-binding motif).

- What is the difference/significance between the oocyte/embryo pictures shown in Fig. 2E and F and Fig. 3G???

Fig 2E and F correspond to transgenic oocytes and embryos expressing GFP-tagged eIF4E1c or eIF4E1b under the control of the β -actin promoter. In contrast, Fig 3G shows wild-type 1k-cell embryos transiently expressing wild-type or mutant versions of eIF4Ea or eIF4E1b by mRNA injections into 1-cell stage embryos. While mRNA injections have allowed us to rapidly test the effect of different mutations on eIF4E1b localization, bypassing the need to generate stable

transgenic lines in zebrafish, the injected mRNAs are not expressed until 3 hours post-fertilization and do not allow us to study earlier developmental time points. Therefore, while both transient and stable expression of eIF4E1b results in the accumulation of eIF4E1b in cytoplasmic granules of zebrafish embryos, the transgenic line has allowed us to image eIF4E1b in oocytes (where there are no distinct foci) and eggs.

- Fig. 3H reports on the relative number of granules relative to wild-type - it would be more helpful to report on the actual number of granules to evaluate the data directly.

As requested by the reviewer, we have now plotted the absolute number of granules in Fig 3H.

- Fig. 3H reports that the cap binding mutant eIF4E1b localizes to granules as efficiently as WT. The authors may want to point out this finding which suggests that the eIF4E1b/P body interaction does not require cap binding/binding to RNA?

We thank the reviewer for this suggestion, and have included an additional sentence in the Results section that highlights this finding, which we also find interesting:

“... mutation of residues located on the dorsal or lateral surface of eIF4E1b that are important for the interaction with eIF4EBP and eIF4ENIF1 abolished its accumulation in granules (Fig 3H, Appendix Figure S3D), indicating that eIF4E1b localization to P-bodies does not require binding to the mRNA cap but to other proteins”.

Fig. 5D and E: the polyA tail length of mRNAs bound by eIF4E1b are shorter on average during early development and become longer with time, eventually reaching average sizes. Because the authors only perform RIP at one stage (8-cell - early development), **whether eIF4E1b continues to bind the mRNAs when they become polyadenylated and translated is not known**. Absent more information (**direct loss of function analyses?**), any conclusion as to the function of eIF4E1b appears premature. In fact the finding that eIF4E1b is required early in female oogenesis suggests an essential function BEFORE the oocyte to embryo transition.

We agree with the reviewer that our previous data did not allow us to draw conclusions about the status of eIF4E1b-bound mRNAs at later time points. We performed RNA immunoprecipitation (RIP) followed by RT-PCR to determine whether transcripts enriched in the eIF4E1b RIP at 1.25 hours post fertilization (hpf) remained bound at 3 hpf. As candidates, we selected four mRNAs enriched in the eIF4E1b RIP at 1.25 hpf (**Fig 1A** below): three of them were reported to increase polyA tail length and TE between 2 and 4 hpf, while one was not (Subtelny et al 2014; **Fig 1B** below). All four transcripts were low abundant in the eIF4E1b RIP at 3 hpf, present at lower levels than in an eIF4E1c RIP and a negative wild-type RIP control (**Fig 1C** below). These data suggest that little or no eIF4E1b molecules are bound to the candidate mRNAs at 3 hpf. Consistent with a release of eIF4E1B during early embryogenesis, it has been reported that mouse eIF4E1B specifically binds to mRNAs in the egg, but immunoprecipitates few mRNAs at the 2-cell embryo stage, a time when zygotic genome activation occurs in mice (Yang et al 2023). Based on this result, Yang et al propose that mRNAs are “no longer bound by eIF4E1b at this stage of development”.

Figure 1. RNA immunoprecipitation (RIP) experiments followed by RT-PCR to assess the possibility of eIF4E1b release from candidate mRNAs during embryogenesis.

A Sequencing reads of four mRNAs immunoprecipitated with eIF4E1b and eIF4E1c in 8-cell embryos at 1.25 hpf. **B** Median polyA tail length (in nucleotides, nt; left) and translation efficiency (TE; right) at 2 and 4 hpf of the mRNAs shown in A (data from Subtelny *et al.*, 2014).

C RT-PCR analyses of mRNAs immunoprecipitated with empty beads (WT, negative control), eIF4E1b and eIF4E1c, and of mRNAs isolated from WT, *actb2:3xflag-sfGFP-eIF4E1b* and *actb2:3xflag-sfGFP-eIF4E1c* embryo lysates at 3 hours post fertilization (hpf; 35 PCR cycles). Expected lengths of the PCR products (in base pairs, bp) according to GRCz11: *eif2s1a* = 189; *esco2* = 156; *wdr74* = 197; *sod1* = 211. WT: wild type.

Moreover, our IP-MS experiments show that eIF4E1b interacts with eIF4ENIF1 and other P-body components at 3 hpf (Dataset EV3). Although eIF4G1a is reliably detected in our 3 hpf MS data, it is not significantly enriched in the eIF4E1b IP. Thus, our IP-MS data suggest that eIF4E1b is not bound to mRNAs that undergo translation at this later time point.

As noted by the reviewer, eIF4E1b function is critical for zebrafish oogenesis. However, the high levels of eIF4E1b during the first hours after fertilization suggest that eIF4E1b can also play a role not only during oogenesis but also during early embryogenesis. In line with this, loss of *eif4e1b* in mice allows oogenesis and leads to an embryonic arrest at the 2-cell stage (Guo *et al.* 2023 and Yang *et al.* 2023). Similar phenotypes have been observed with Zar1: while a double *zar1*, *zar2* knockout mouse arrests at the 1- or 2-cell stage (Rong *et al.* 2019), loss of Zar1 in zebrafish impairs oogenesis and causes sex reversal (Miao *et al.* 2017).

Referee #2:

In this manuscript, Lorenzo-Orts *et al.* aim to uncover how maternal mRNAs, stored during oogenesis with short polyA tails, avoid degradation.

The scientific premise pointing to eIF4Es as protein candidates to be in charge of the maternal mRNA dormancy is speculative, but the authors zero in the eIF4E1b due to its high expression during oogenesis and in the early embryo.

Using an elegant CRISPR/Cas9 strategy, the authors show that upon loss of eIF4e1b, there is impaired oocyte development and sex reversal - no females are formed. The few female-looking fish have no oocytes but a homogeneous cell mass.

The authors demonstrate that eIF4E1b from mouse interacts with the cap structure of mRNAs and with the proteins eIF4ENIF1 and eIF4EBP1, but no eIF4G, excluding a role for eIF4E1b in promoting translation. This data contradicts previous publications on eIF4E1b but the confidence in these results is based on the fact that eIF4E1b contains the key aminoacids that mediate these interactions.

Next the authors conduct immunoprecipitation of eIF4E1b from transgenic embryos and identify the partners by mass spectrometry. This data reveals that eIF4E1b interacts with many p-body components but not the decapping enzymes, suggesting that eIF4E1b target maternal mRNAs to p-bodies yet prevents their decapping and degradation.

To identify the targets of eIF4E1b, the authors conduct a RIP-Seq experiment. It reveals that the mRNA targets of eIF4E1b, are stabilized by the interaction and that are transcriptionally repressed, despite they show elongation of the polyA tail.

Overall, this is a rigorous work showcasing eIF4E1b as a novel factor involved in the translation repression and stability of maternal mRNAs. The data is well presented and the experimental strategy is rigorous, which includes establishing several zebrafish transgenic and mutant lines. The results from this work will be the seed for future studies aiming at understanding long-term stability of maternal mRNAs.

We thank the reviewer for the positive feedback.

Therefore, the manuscript is adequate for publication in EMBO Reports if the authors address the following question: The overarching goal of the manuscript is to uncover the factor responsible for the dormancy of maternal mRNA. While indeed the authors show how eIF4E1b binds to capped mRNAs and prevents their translation by localization to P-bodies and their decay by not interacting with decapping enzymes, **it is no clear if this regulation mediated by eIF4E1b is dynamic and developmentally regulated**. It would be expected that the factor involved in silencing maternal mRNAs would **relieve its repression at some point during early embryogenesis** (e.g. onset of zygotic transcription, maternal-to-zygotic transition) to ensure normal development. The authors base their analysis in samples during late oogenesis or at 8-cell stage embryos. Figure 1D shows how the eIF4E1b protein levels decay by shield stage. **The authors should determine if eIF4E1b losses affinity for mRNA at later stages of embryogenesis, and if the mRNAs associated with eIF4E1b eventually recover translation potential**.

We thank the reviewer for raising these very interesting points.

Using our transgenic line expressing *actb2:3xflag-GFP-eIF4E1b*, we have now performed RIP followed by RT-PCR to investigate whether mRNAs bound by eIF4E1b in 8-cell stage embryos remain bound to eIF4E1b at 3 hours post-fertilization (hpf), a time when the median polyA tail length increases. In addition, we have also used eIF4E1c and “empty” beads (preincubated with wild-type lysates) as baits to assess whether mRNAs switch to a different eIF4E paralog during embryogenesis and to control for non-specific binding to the anti-GFP beads, respectively. As candidates for RT-PCR, we selected four mRNAs enriched in the eIF4E1b RIP at 1.25 hpf (**Fig 1A** below), three of which show an increase in polyA tail length and translation efficiency between 2 and 4 hpf (Subtelny et al 2014; **Fig 1B** below).

Although we were able to amplify all candidate transcripts from total embryo lysates (inputs), the abundance of these transcripts was lower in the eIF4E1b RIP than in the eIF4E1c RIP and the wild-type RIP, suggesting that little or no eIF4E1b molecules bind to the selected transcripts at 3 hpf (**Fig 1C** below). Consistent with a release of eIF4E1B during early embryogenesis, Yang et al proposed that mouse eIF4E1B loses its binding to mRNAs at the 2-cell stage, when zygotic genome activation occurs in mice (Yang et al 2023).

Figure 1. RNA immunoprecipitation (RIP) experiments followed by RT-PCR to assess the possibility of eIF4E1b release from candidate mRNAs during embryogenesis.

A Sequencing reads of four mRNAs immunoprecipitated with eIF4E1b and eIF4E1c in 8-cell embryos at 1.25 hpf. **B** Median polyA tail length (in nucleotides, nt; left) and translation efficiency (TE; right) at 2 and 4 hpf of the mRNAs shown in A (data from Subtelny *et al*, 2014).

C RT-PCR analyses of mRNAs immunoprecipitated with empty beads (WT, negative control), eIF4E1b and eIF4E1c, and of mRNAs isolated from WT, *actb2:3xflag-sfGFP-eIF4E1b* and *actb2:3xflag-sfGFP-eIF4E1c* embryo lysates at 3 hpf (35 PCR cycles). Expected lengths of the PCR products (in base pairs, bp) according to GRCz11: *eif2s1a* = 189; *esco2* = 156; *wdr74* = 197; *sod1* = 211. WT: wild type.

Moreover, regarding the reviewer's question about whether mRNAs associated with eIF4E1b regain translation potential, based on published data (Subtelny *et al* 2014), the translational efficiency of eIF4E1b RIP targets increases during embryogenesis (Fig 5E). However, our MS data support the idea that eIF4E1b is not involved in translation initiation even at later developmental time points, as P-body components, but not eIF4Gs, were specifically enriched in the 3 hpf eIF4E1b IP (Dataset EV3). Therefore, our data suggest that eIF4E1b-bound transcripts must be released for either translation or degradation during embryogenesis. While we find it unlikely that eIF4E1b stores maternal mRNAs only for later degradation in the embryo, we lack direct experimental evidence to address this non-trivial question. Potential experiments, such as single-molecule tracking of eIF4E1b-bound mRNAs during embryogenesis to follow the fate of both the protein and the mRNA, will require considerable time and effort and may open new avenues of research beyond the scope of this paper.

To investigate whether mRNAs are released from eIF4E1b during embryogenesis, we also considered potential mechanisms that might trigger this release. To our knowledge, the only mechanism reported so far to reduce eIF4E's affinity for the mRNA cap is phosphorylation of Ser209 in human eIF4E (Scheper et al 2002; Zuberek et al 2003; Slepnev et al 2006). Since this residue is conserved in eIF4E1Bs, we tested whether a phosphomimic version of human eIF4E1B would have reduced affinity for the mRNA cap. However, we observed similar binding to the mRNA cap for wild-type, phosphomimic and phosphomutant versions of human eIF4E1B (see **Figure 2** below), providing no evidence that phosphorylation triggers the release of eIF4E1b from dormant mRNAs.

Figure 2. Pulldown assays to test the effect of Ser234 phosphorylation on human eIF4E1B binding to the mRNA cap. Coomassie-stained gel of wild-type, phosphomutant (S to A) and phosphomimic (S to D) versions of human eIF4E1B proteins immunoprecipitated with m⁷G-coated beads (left). Quantification of 3 different experiments is shown on the right.

Alternatively, eIF4E1b may be outcompeted by other eIF4Es, such as eIF4E1c, whose levels increase at around 2 hpf. Another possibility is that the release of eIF4E1b may be triggered by a change in the eIF4E1b protein interactome. For instance, Zar1l is progressively degraded after fertilization but still detected by TMT-MS at 3 hpf (Fig EV4E). While Zar1l strongly associates with eIF4E1b in oocytes and early embryos, it is not detected in the eIF4E1b IP-MS at 3 hpf (Fig 4A-C). Further research is needed to understand the mechanism by which factors that have been implicated in maternal mRNA dormancy and that interact with eIF4E1b in the presence of mRNA, such as Zar1, Zar1l and Ybx1, contribute to the selection and release of dormant mRNAs. We hope that the reviewer understands that these questions, while very exciting, are beyond the scope of this paper.

Referee #3:

Lorenzo-Ortiz, L et al. in this manuscript discuss a role for eIF4E1B in the stability and translation repression of maternally deposited mRNAs in zebrafish. First, the authors generated KO zebrafish lines and showed that eIF4E1B also plays a role in ovary development, leading to male bias in homozygous KO fish (Fig 1). Then, the authors tested several eIFs and their capability to bind to mRNA's cap and other eIFs (Fig 2). Since eIF4E1B interacts with eIF4ENIF1, known to localize to P-bodies, the authors show through microscopy puncta expression of eIF4E1B but not of other eIFs (Fig 2). To explore such protein interactions, the authors performed several binding predictions, that were later validated in vitro and in vivo with mutant versions of eIF4E binding motifs, showing loss of puncta expression in injected zebrafish embryos (Fig 3). Further, authors

looked for the protein interactome of eIF4E1B using IP followed by mass spectrometry and found enrichment in P-body components, but not of translation factors (Fig 4). Finally, authors employed RNA immunoprecipitation and identified that eIF4E1B binds to maternal mRNAs with short polyA tail hinting its function as prevention of decay and translation (based on RNA-seq and ribosome profiling datasets) (Fig 5).

Major:

Overall, the manuscript is well written, and figures correctly address questions posed by the authors. In addition, authors elegantly show eIF4E1B protein interaction with other eIFs and generated IP datasets will be valuable resources for others interested in studying translation regulation in early zebrafish embryogenesis.

We thank the reviewer for the positive comments on our manuscript.

Nevertheless, it was recently shown in mice that deletion of eIF4E1B causes arrest at the 2-cell stage (Yang, G. et al. 2023 - Genes & Development; Guo, J. et al., 2023 - Advanced Science) (both cited in the manuscript), **yet the authors do not provide results or discussion if a similar arrest in development happens in the generated zebrafish knock-out lines.** If eIF4E1B is instrumental for maternal mRNA dormancy and protection in zebrafish, then **I would expect that the loss of function mutants show early developmental problems either before or at the onset of genome activation.**

We would like to clarify to the reviewer that in zebrafish, *eif4e1b* mutant females do not develop oocytes, and thus we cannot obtain *eif4e1b* mutant zebrafish embryos depleted of maternal eIF4E1b. As the reviewer correctly points out, loss of eIF4E1B in mice causes embryonic arrest at the 2-cell stage. However, Guo et al. show that ovaries of *eif4e1b* knockout mice also display severe morphological defects. In particular, *eif4e1b* mutant mouse ovaries are larger and contain more follicles than wild-type ovaries. Therefore, according to the phenotype reported by Guo et al., mouse eIF4E1B may also play a role during oogenesis in mice. An oogenesis defect upon loss of eIF4E1B is consistent with a role for this protein in regulating maternal mRNA dormancy, since maternal mRNAs are transcribed during oogenesis and their dormancy is established prior to fertilization.

Differences in severity and timing of phenotypes between mice and zebrafish are not uncommon and have also been reported for other factors involved in maternal mRNA storage and repression, such as Zar1 and Zar1l (also known as Zar2). Loss of Zar1 and Zar2 in mice leads to embryonic arrest at the 1- or 2-cell stage (Rong et al 2019), whereas loss of Zar1 in zebrafish is sufficient to impair oogenesis and cause sex reversal (Miao et al 2017). Therefore, the phenotype reported for eIF4E1b in mouse and zebrafish is consistent with that reported for the loss of a dormancy factor, supporting a role for eIF4E1b in this process.

Moreover, authors imply a major role of eIF4E1B in the stability of maternal mRNAs with short poly-A tail, yet there is **no data showing that the absence of eIF4E1B causes decay of these mRNAs.** Most of the data raise from RIP-seq experiments follow by the characterization of targets. In other words, eIF4E1B might **interact with genes depending in the poly(A)-tail length** but might **not have any role during early development.** To show that those correlation are actually part of the eIF4E1B, it would be nice to **measure poly(A)-tail length, translation (Ribosome profiling) and mRNA decay** profiles in the mutant and wild-type embryos. Indeed, if such an effect

is not achieved in zebrafish it might be because **eIF4E1B is not as instrumental in zebrafish as it is in mice or it might be redundancy problem. Could it be that in mice the protein is not as highly deposited as in zebrafish?**

We agree with the reviewer that obtaining direct data on eIF4E1b's ability to repress translation and stabilize mRNAs would strengthen our conclusions. Functional assays to directly assess the function of eIF4E1b are challenging because loss of eIF4E1b in zebrafish disrupts oogenesis and causes sex reversal, so it is not possible to obtain homozygous *eif4e1b* females and therefore embryos lacking eIF4E1b. Although we can obtain homozygous *eif4e1b* embryos in a 1:4 ratio by crossing heterozygous fish, these embryos contain maternally deposited eIF4E1b mRNA and protein and therefore cannot be used to study eIF4E1b loss during early embryogenesis.

We agree with the reviewer that measuring the polyA tail length and the translational efficiency of eIF4E1b-bound transcripts would strengthen our conclusions. However, the experiments suggested by the reviewer are not trivial as they require much more input than the amount of mRNA we recover by RIP. We have performed additional experiments that allow us to evaluate the effect of eIF4E1b on mRNA stability and translation:

1) **Proteomic analysis of *eIF4E1b* overexpressing embryos (Fig 6A-F).** Overexpression of *eIF4E1b* with the β -actin promoter results in the downregulation of a large number of proteins at 5 hours post fertilization, in contrast to the large number of proteins upregulated by eIF4E1c overexpression. For some of them, we show by qPCR that the downregulation is not at the mRNA level, suggesting that eIF4E1b interferes with the translation of specific transcripts.

2) **Proteomics and transcriptomics on female gonads of *eif4e1b* mutant and wild-type juvenile fish with a high expression of *ziwi:GFP* (Fig 6G-I and Appendix Figure S4).** We used our *ziwi:GFP* transgenic line to isolate female gonads from *eif4e1b* mutant and wild-type juveniles before *eif4e1b* mutants revert to males. Mass spectrometry data revealed considerably more up- than down-regulated proteins in *eif4e1b* mutant gonads, the majority of which were not up-regulated at the mRNA level. This is in line with an increased translation in the absence of eIF4E1b. Moreover, many mRNAs that were enriched in the eIF4E1b RIP at 1.25 hpf were depleted in *eif4e1b* mutant gonads, such as histone mRNAs, consistent with mRNA destabilization upon eIF4E1b loss.

3) **Tethering eIF4E1b or eIF4Ea to a reporter mRNA (Fig 5F-I).** Using the MS2/MCP system, we tethered either canonical eIF4Ea or eIF4E1b to a *GFP* reporter mRNA in the zebrafish embryo. We artificially tethered these proteins to the 3' UTR of a reporter mRNA because (1) MS2 loops in the 5' UTR impair translation in our hands, and (2) eIF4G binding to the 3' UTR has been shown to promote translation (Paek et al 2015). Our results show that, in contrast to canonical eIF4Ea, eIF4E1b was not able to promote the translation of the *GFP* mRNA.

Importantly, we would like to point out that the phenotype we observe in zebrafish upon loss of eIF4E1b is, if anything, stronger than that reported in mice. While it is possible to obtain *eif4e1b* knockout females in mice, only a small percentage of zebrafish develop into females with dysfunctional gonads in adulthood. However, since female mice cannot reverse their sex if oogenesis is not functioning properly, we think it is unfair to make any claims about the strength of the phenotypes. Nevertheless, we can confidently conclude that eIF4E1b is essential for egg development in zebrafish.

While it is very attractive to propose the eIF4E1B play a role in maternal mRNA dormancy and be part "fundamental post-transcriptional regulatory principles governing early vertebrate development" (abstract). Based on the results (phenotypes), it looks that eIF4E1B play a fundamental role in female germline development in zebrafish. Maybe this should be the title of the work.

We hope that our response above clarifies to the reviewer that the phenotype we observe in our *eif4e1b* mutant is consistent with that expected from the loss of a factor involved in maternal mRNA dormancy in zebrafish. While we thank to the reviewer for the title suggestion, we believe that our manuscript provides data beyond the phenotypic characterization of an *eif4e1b* mutant in zebrafish.

Minor:

1) In Figure 1 the authors claim that eIF4E1B is highly expressed in zebrafish oogenesis and early embryogenesis, but panels C and D only show expression levels of selected eIF4Es in comparison to each other. How highly expressed is eIF4E1B when compared to the whole transcriptome and proteome? Can the authors add a reference line of for example the median of the total transcriptome and proteome for a better comparison?

In response to the reviewer's suggestion, we have added dashed lines indicating the mean and median of transcript and protein levels during zebrafish oogenesis and/or embryogenesis in Fig 1C and D.

2) There is a typo in line number 360: "were shacked at 800 rpm", which should be "shaken" (verb for motion).

We thank the reviewer for pointing out this typo, which we have now corrected in the revised version of the manuscript.

3) Supplemental excel tables lack description about their contents and how they are organized (meaning of colors, what the summary sheet stands for, etc).

We thank the reviewer for pointing this out. We have included descriptive legends in a separate tab of each file, as is standard for EMBO Reports.

4) As a dataset resource, it would be ideal to add o with all identified proteins from mass spectrometry proteomics data not filtered by p-value of FC analysis performed in the tables.

We have now added the non-significant hits identified by MS in separate tabs.

Dear Andrea,

Thank you for the submission of your revised manuscript. We have now received the enclosed reports from the referees that were asked to assess it. Both referees still have a few more minor suggestions that I would like you to address and incorporate before we can proceed with the official acceptance of your manuscript. Please also send us a point-by-point response to all last comments.

A few editorial requests will also need to be addressed:

- Please mention all species used in this study in the abstract.
- In the author list, Friederike Leesch is listed in the ms vs. Katrin Leesch in our online submission system, please correct.
- Please remove the author credits from the ms file. All credits need to be entered online during ms submission.
- Some FUNDING INFO is missing in our online submission system, please add.
- The Dataset EV5 needs correction of its label, it says Dataset EV2 in the file.
- Please address these comments by our data editors:

1. Please indicate the statistical test used for data analysis in the legends of figures 4a-f; 5a; 6a-b, g-h; EV4a-c; EV5c
2. Please note that information related to n is missing in the legends of figures 4a-f; 5a, c; 6a-c, g-h; EV2e-f; EV4a-c; EV5c.

- During our routine image analysis of ms to be accepted, we detected very similar WB bands, possibly identical:

1. Possible re-use of westernblot. Figure 2D Dr eIF4Ea and Appendix Fig S3B eIF4E1Bdorsal(W84A). No Source data for appendix to check this.
2. Appendix figure S3B - Mm eIF4EN-eIF4E1BC and Mm eIF4E1BN-eIF4EC are very similar. possibly different but not noticeably.

Can you please clarify and send us the source data for Appendix figure S3B?

- Please modify the title and abstract, as suggested by the referees.

EMBO press papers are accompanied online by A) a short (1-2 sentences) summary of the findings and their significance, B) 2-3 bullet points highlighting key results and C) a synopsis image that is exactly 550 pixels wide and 200-600 pixels high (the height is variable). You can either show a model or key data in the synopsis image. Please note that text needs to be readable at the final size. Please send us this information along with the final manuscript.

Best,
Esther

Referee #1:

The main criticism raised in the first round of reviews is that the authors provided no direct evidence to support the hypothesis that eIF4E1b functions as a "dormancy factor" : a factor essential to maintain maternal mRNAs translationally repressed in oocytes and early embryos but competent for translational activation later in embryos.

The authors have added new data that supports the idea that, unlike canonical cap binding proteins, eIF4E1b can represses (rather than activate) translation, but these data do not yet demonstrate a requirement for eIF4E1b as a "dormancy factor".

Overall we recommend publication provided the authors modify statements such as " Here, we report that the germline eIF4E paralog eIF4E1B, which is conserved in vertebrates, is involved in maternal mRNA dormancy by binding to the cap mRNAs that are reported to have short or no polyA tails, thereby contributing to their stability and translational repression" . Such statements should be clearly presented as HYPOTHESES rather than firm conclusions. Also, the idea that some cap binding proteins repress, rather than stimulate, translation has already been proposed based on experiments in *C. elegans*, and this literature should be better referenced by the authors (see point #1).

1. The authors still do not cite accurately Huggins et al., 2020 <https://pubmed.ncbi.nlm.nih.gov/32079657/> where it is shown that *C. elegans* IFE-3, an eIF4E1b paralog, promotes oogenesis and represses the translation of maternal mRNAs as the authors suggest in this study for eIF4E1b. Also another study reported no interaction between IFE-3 and translation initiation factors <https://genesdev.cshlp.org/content/33/13-14/857.full>. This seems very relevant to the work presented here.

2. The authors now include new gain of function experiments (Fig 5F-I, Fig 6) that add supporting evidence for the role of eIF4E1b as a translation repressor.

Figure 5F-I shows an MS2 tethering experiment where eIF4E1b was tethered to the 3'UTR of GFP reporter transcript in vivo and shown to repress GFP expression. The value of this experiment is dubious given that eIF4E1b normally binds mRNAs via the CAP and not via the UTR. Also, the negative control eIF4Ea caused a decrease of GFP expression, although the authors argue that this was related to its effect on mRNA degradation. Overall this experiment seems difficult to interpret.

Figure 6A-B reports on overexpression experiments where eIF4E1b or eIF4E1c were overexpressed 5hpf zebrafish embryos and protein levels were measured. Satisfyingly, the majority of proteins dysregulated in the eIF4E1b overexpression were down regulated (down in eIF4E1b vs WT). This is a very nice experiment that suggest a repressive role for eIF4E1b but does not demonstrate a REQUIREMENT for eIF4E1b in translationally repressing and storing maternal mRNAs.

Figure 6D-F presents RNAseq and ribosome-protected-fragment sequencing data from Subtelny et al and shows calculated translational efficiencies for the transcripts of proteins dysregulated in 6A-B. The authors claim these data show that there are no gross changes in translational efficiency or RNA levels across these groups compared to the median of all the RNAs detected by Subtelny et al. The authors do not state if these data were collected from embryos at a similar timepoint as their MS experiments. These data provide correlative evidence to suggest excess eIF4E1b reduces the translation of at least 331 proteins in the embryo when overexpressed.

Figure 6G-I shows the result of a loss of function experiment where the authors carried out MS/MS and RNAseq on eIF4E1b^{-/-} early female gonads. The authors identify 1229 proteins overexpressed in eIF4E1b^{-/-} fish compared to wild-type, and a smaller number of underexpressed proteins (219) including P body factors known to be required for female gonad differentiation. They overlap their MS data with RNAseq data, but only highlight the down regulation of many histone transcripts. Venn diagrams of up versus down regulated proteins and RNAs are presented, but limited qualitative information is provided about these overlaps. These data are consistent with an important role for eIF4E1b in growing oocytes, but these data do not address a proposed role in "maternal mRNA dormancy" since such a role would presumably manifest at later stages of oogenesis/early embryogenesis. In conclusion, the authors provide new experiments (1. gain-of-function evidence that eIF4E1b can function as a translational repressor and 2. loss of function evidence that eIF4E1b regulates gene expression in growing oocytes) that supports the view that eIF4E1b does not function as canonical translational activator. The authors, however, still lack direct evidence to demonstrate a role for eIF4E1b specifically for "maternal mRNA dormancy" in mature eggs/early embryos as indicated in the title. We suggest the authors modify the title, abstract, and discussion to focus on the role of eIF4E1b more broadly as a putative translational repressor as also supported by prior evidence in Huggins et al., 2020.

Referee #2:

In this revised version of the manuscript, Lorenzo-Orts et al., aim to address my previous comment about if the regulation mediated by eIF4E1b is dynamic and regulated during embryogenesis. In the initial version of the manuscript, the authors clearly demonstrate that eIF4E1b binds to target mRNA to prevent both their translation and degradation. However, it was not clear if that double regulation served a developmental purpose, as little data was shown about the outcome of the target mRNAs at later stages of embryogenesis.

In the current manuscript, the authors have performed a RIP experiments with eIF4E1b and controls to demonstrate that eIF4E1b stop interacting with its target mRNAs by 3 hours post-fertilization. This result speak for the dynamic nature of eIF4E1b regulation and its involvement in early embryogenesis.

Once it is understood that eIF4E1b no longer interacts with its mRNA targets at later stages of development, the authors set to address if this lack of interaction is due to mRNA degradation or release of the mRNAs from eIF4E1b. The authors tested if a phosphorylation event could modify the affinity of eIF4E1b for mRNA but they could not determine significant differences in the conditions tested.

Collectively, the work presented here to address my question and the ones raised by the other reviewers expand the interest of eIF4E1b function to later stages of vertebrate embryogenesis and merits publication on EMBO Reports, as this work will serve as steppingstone for more analysis about translation regulation in the field.

The authors could improve the final version of the manuscript addressing the following points:

1.- The RIP experiment that shows that eiF4E1b does not bind as many mRNAs as eiF4E1c at 3 hpf. This experiment also shows as a control a RIP experiment performed in WT embryos not expressing any FLAG protein, yet still shows RNA binding. The authors should clarify why they observe higher background binding in the control beads than with eiF4E1b. One possible reason is that the binding of eiF4E1b to the beads may block non-specific RNA binding.

2.- In their reply to my comment, the authors show that targets of eiF4E1b are re-adenylated at later stages of embryogenesis, yet it is not clear if the targets are released from eiF4E1b and re-adenylated or just degraded. The authors recently published a compendium of SLAM-seq data on early zebrafish embryogenesis that would help to determine if the bulk of eiF4E1b targets present at later stages are maternally provided or zygotically transcribed. If most of the signal comes from maternally provided genes, it will support the hypothesis that eiF4E1b releases its targets, allowing their translation.

Response to reviewer comments:

We again thank the reviewers for their time and effort in reviewing our manuscript. Below we provide point-by-point responses to their comments. Reviewer comments are in **blue**, and responses are in **black**.

Referee #1:

The main criticism raised in the first round of reviews is that the authors provided no direct evidence to support the hypothesis that eIF4E1b functions as a "dormancy factor" : a factor essential to maintain maternal mRNAs translationally repressed in oocytes and early embryos but competent for translational activation later in embryos.

The authors have added new data that supports the idea that, unlike canonical cap binding proteins, eIF4E1b can represses (rather than activate) translation, but these data do not yet demonstrate a requirement for eIF4E1b as a "dormancy factor". Overall we recommend publication provided the authors modify statements such as " Here, we report that the germline eIF4E paralog eIF4E1B, which is conserved in vertebrates, is involved in maternal mRNA dormancy by binding to the cap mRNAs that are reported to have short or no polyA tails, thereby contributing to their stability and translational repression" . Such statements should be clearly presented as HYPOTHESES rather than firm conclusions. Also, the idea that some cap binding proteins repress, rather than stimulate, translation has already been proposed based on experiments in *C. elegans*, and this literature should be better referenced by the authors (see point #1).

1. The authors still do not cite accurately Huggins et al., 2020 <https://pubmed.ncbi.nlm.nih.gov/32079657/> where it is shown that *C. elegans* IFE-3, an eIF4E1b paralog, promotes oogenesis and represses the translation of maternal mRNAs as the authors suggest in this study for eIF4E1b. Also another study reported no interaction between IFE-3 and translation initiation factors <https://genesdev.cshlp.org/content/33/13-14/857.full>. This seems very relevant to the work presented here.

We thank the reviewer for pointing out these studies in *C. elegans*. Based on the reviewer's comments, we investigated the similarities between vertebrate and worm eIF4Es. To this end, we extended the phylogenetic analysis of eIF4E proteins shown in Fig S1 to eukaryotes, including the five *C. elegans* eIF4E paralogs IFE-1 to IFE-5 (new **Appendix Fig S5**). While one IFE (IFE-4) is an eIF4E2 ortholog, the other four are closer to class I eIF4Es (which includes canonical eIF4E and eIF4E1b), but do not cluster together with eIF4Es or eIF4E1bs. Therefore, based on conservation, we could not identify a clear eIF4E1b ortholog in nematodes.

However, the reviewer is correct in pointing out that IFE-3 shares some functional similarities with eIF4E1b, since of the five IFES present in *C. elegans*, only IFE-3 is essential for oogenesis (Huggins et al 2020) and embryogenesis (Keiper et al. 2000) and interacts with the worm eIF4ENIF1 ortholog IFET-1 in vitro and in vivo. IFE-3 has been implicated in the repression of specific mRNAs, such as *fem-3*, *daz-1* and *mex-3* (Huggins et al. 2020; Mennatallah et al. 2021). However, *ife-3* downregulation (RNAi) results in less polysomes in worm extracts (Huggins et al. 2020), consistent with IFE-3 promoting rather than repressing translation. Based on this finding, the authors of this study conclude: "Notably, IFE-3 is the nematode ortholog of canonical eIF4E-1, and would be expected to exert positive translational activity on some population of mRNAs, consistent with diminished polysome content following RNAi. Intriguingly, our polysome data also infer that IFET-1 (the nematode ortholog of human eIF4E-nuclear transport protein) also assists in that positive role". Therefore, whether IFE-3 is able to promote translation remains unclear.

While Cordeiro Rodrigues et al. do not detect translation initiation factors associated with IFE-3, other studies show that IFE-3 can bind to the worm eIF4G ortholog IFG-1 in vitro (Peter et al. 2015, Fig 6F). This result suggests that unlike eIF4E1b, which does not interact with eIF4G, *C. elegans* IFE-3 is capable of forming the eIF4F complex required for cap-dependent translation. Although eIF4E1b shares functional similarities with IFE-3, these data suggest that they may not be fully equivalent. Moreover, the differences between zebrafish and *C. elegans* (in terms of biology and eIF4E protein conservation) make it difficult to draw any conclusions. Nevertheless, we have acknowledged previous studies on IFE-3 in the *Discussion* of our manuscript, which now reads:

“The *C. elegans* eIF4E paralog IFE-3 (Appendix Fig S5) has also been reported to interact with the worm eIF4ENIF1 ortholog IFET-1 in the germline and to be important for oocyte cell fate (Huggins et al, 2020) and embryonic development (Keiper et al, 2000). We propose that eIF4ENIF1-mediated repression of eIF4E1b-bound maternal mRNAs in P-bodies is also important for vertebrate oogenesis and embryogenesis”.

2. The authors now include new gain of function experiments (Fig 5F-I, Fig 6) that add supporting evidence for the role of eIF4E1b as a translation repressor.

Figure 5F-I shows an MS2 tethering experiment where eIF4E1b was tethered to the 3'UTR of GFP reporter transcript in vivo and shown to repress GFP expression. The value of this experiment is dubious given that eIF4E1b normally binds mRNAs via the CAP and not via the UTR. Also, the negative control eIF4Ea caused a decrease of GFP expression, although the authors argue that this was related to its effect on mRNA degradation. Overall this experiment seems difficult to interpret.

As the reviewer points out, the tethering assays performed to assess the effect of eIF4Es on an mRNA are artificial because eIF4Es normally bind to the mRNA cap and not to the 3' UTR. However, tethering eIF4Es to the mRNA cap is challenging because we would need to ensure that the proteins remain bound to the cap of the reporter mRNA after injection into 1-cell stage embryos. Since tethering eIF4G to the 3' UTR has been reported to increase translation of a reporter mRNA (Paek et al, 2015), we decided to use a similar strategy.

Tethering eIF4E1b to the 3' UTR of a *gfp* reporter mRNA resulted in a 60% reduction in GFP levels, while tethering eIF4Ea resulted in a smaller reduction (24% decrease). However, we also observed the almost complete degradation of the reporter mRNA after eIF4Ea tethering. Although *gfp* mRNA levels were higher after eIF4E1b binding than after eIF4Ea binding, GFP protein levels were higher when the mRNA was bound by eIF4Ea, consistent with eIF4Ea promoting translation of the small amount of reporter mRNA remaining after eIF4Ea expression. The comparison of protein and mRNA levels of a reporter mRNA has been used in other studies to estimate translation (*translation quotient*; Rahaman et al. 2023), and we find it a fairer estimate than looking at the protein product alone.

While we agree that the likely artificial destabilization of the reporter mRNA observed upon eIF4Ea tethering makes our results not straightforward to interpret, the striking effects observed after eIF4Ea and eIF4E1b binding suggest major functional differences between these two proteins that are consistent with eIF4E1b not promoting translation, as opposed to canonical eIF4Ea.

Figure 6A-B reports on overexpression experiments where eIF4E1b or eIF4E1c were overexpressed 5hpf zebrafish embryos and protein levels were measured. Satisfyingly, the majority of proteins dysregulated in the eIF4E1b overexpression were down regulated (down in eIF4E1b vs WT). This is a very nice experiment that suggest a repressive role for eIF4E1b but does not demonstrate a REQUIREMENT for eIF4E1b in translationally repressing and storing maternal mRNAs.

We thank the reviewer for the positive feedback on this experiment. Since maternal mRNA dormancy occurs earlier in development and eIF4E1b is required for zebrafish oogenesis, we agree with the

reviewer that this experiment does not allow to draw conclusions about the requirement of eIF4E1b for maternal mRNA dormancy, but is consistent with eIF4E1b repressing translation.

Figure 6D-F presents RNAseq and ribosome-protected-fragment sequencing data from Subtelny et al and shows calculated translational efficiencies for the transcripts of proteins dysregulated in 6A-B. The authors claim these data show that there are no gross changes in translational efficiency or RNA levels across these groups compared to the median of all the RNAs detected by Subtelny et al. The authors do not state if these data were collected from embryos at a similar timepoint as their MS experiments. These data provide correlative evidence to suggest excess eIF4E1b reduces the translation of at least 331 proteins in the embryo when overexpressed.

The data from Subtelny et al. was obtained from embryos at 4 hours post fertilization, as stated in the figure legend.

Figure 6G-I shows the result of a loss of function experiment where the authors carried out MS/MS and RNAseq on eIF4E1b^{-/-} early female gonads. The authors identify 1229 proteins overexpressed in eIF4E1b^{-/-} fish compared to wild-type, and a smaller number of underexpressed proteins (219) including P body factors known to be required for female gonad differentiation. They overlap their MS data with RNAseq data, but only highlight the down regulation of many histone transcripts. Venn diagrams of up versus down regulated proteins and RNAs are presented, but limited qualitative information is provided about these overlaps. These data are consistent with an important role for eIF4E1b in growing oocytes, but these data do not address a proposed role in "maternal mRNA dormancy" since **such a role would presumably manifest at later stages of oogenesis/early embryogenesis.**

As mentioned by the reviewer above, we should consider that a factor implicated in maternal mRNA dormancy must be involved in the transient repression of maternal mRNAs during development. However, this repression may not necessarily be established at late stages of oogenesis or early embryogenesis, as some mRNAs may need to be repressed early. For example, histone mRNAs are transcribed early in oogenesis (Cabrera-Quio et al. 2021), but may need to be repressed in stage I oocytes as the rest of oogenesis proceeds in the absence of cell division. eIF4E1b may already play a role in maternal mRNA dormancy at early stages of oogenesis given the *eif4e1b* mutant phenotype and the molecular phenotype observed in early *eif4e1b* mutant gonads. Our data show that histone mRNAs are depleted from early *eif4e1b* mutant gonads. Since the translation of histone mRNAs is coupled to their decay, our results suggest that eIF4E1b contributes to the repression and stabilization of histone mRNAs, whose translation may only be required later in embryogenesis.

In conclusion, the authors provide new experiments (1. gain-of-function evidence that eIF4E1b can function as a translational repressor and 2. loss of function evidence that eIF4E1b regulates gene expression in growing oocytes) that supports the view that eIF4E1b does not function as canonical translational activator. The authors, however, still lack direct evidence to demonstrate a role for eIF4E1b specifically for "maternal mRNA dormancy" in mature eggs/early embryos as indicated in the title. We suggest the authors modify the title, abstract, and discussion to focus on the role of eIF4E1b more broadly as a putative translational repressor as also supported by prior evidence in Huggins et al., 2020.

In light of our response above, we agree with the reviewer that our data do not allow us to conclude that eIF4E1b is required for maternal mRNA dormancy because we do not know the specific time in development at which dormancy is established for each transcript, as we lack ribosome profiling data during zebrafish oogenesis. However, we can confidently conclude that eIF4E1b regulates maternal mRNA dormancy, as eIF4E1b binds to maternal mRNAs that are reported to be deadenylated and poorly translated in the early embryo (i.e. dormant) and its loss or overexpression cause strong molecular phenotypes.

We have removed claims about the requirement of eIF4E1b for maternal mRNA dormancy and modified the title and abstract, which now read:

Title: "eIF4E1b is a non-canonical eIF4E regulating maternal mRNA dormancy".

Abstract: "Maternal mRNAs are essential for protein synthesis during oogenesis and early embryogenesis. To adapt translation to specific needs during development, maternal mRNAs are translationally repressed by shortening the polyA tails. While mRNA deadenylation is associated with decapping and degradation in somatic cells, maternal mRNAs with short polyA tails are stable. Here we report that the germline-specific eIF4E paralog, eIF4E1b, is essential for zebrafish oogenesis. eIF4E1b localizes to P-bodies in zebrafish embryos and binds to mRNAs with reported short or no polyA tails, including histone mRNAs. Loss of eIF4E1b results in reduced histone mRNA levels in early gonads, consistent with a role in mRNA storage. Using mouse and human eIF4E1Bs (in vitro) and zebrafish eIF4E1b (in vivo), we show that unlike canonical eIF4Es, eIF4E1b does not interact with eIF4G to initiate translation. Instead, eIF4E1b interacts with the translational repressor eIF4ENIF1, which is required for eIF4E1b localization to P-bodies. Our study is consistent with an important role of eIF4E1b in regulating mRNA dormancy and provides new insights into fundamental post-transcriptional regulatory principles governing early vertebrate development".

Referee #2:

In this revised version of the manuscript, Lorenzo-Orts et al., aim to address my previous comment about if the regulation mediated by eIF4E1b is dynamic and regulated during embryogenesis. In the initial version of the manuscript, the authors clearly demonstrate that eIF4E1b binds to target mRNA to prevent both their translation and degradation. However, it was not clear if that double regulation served a developmental purpose, as little data was shown about the outcome of the target mRNAs at later stages of embryogenesis.

In the current manuscript, the authors have performed a RIP experiments with eIF4E1b and controls to demonstrate that eIF4E1b stop interacting with its target mRNAs by 3 hours post-fertilization. This result speak for the dynamic nature of eIF4E1b regulation and its involvement in early embryogenesis.

Once it is understood that eIF4E1b no longer interacts with its mRNA targets at later stages of development, the authors set to address if this lack of interaction is due to mRNA degradation or release of the mRNAs from eIF4E1b. The authors tested if a phosphorylation event could modify the affinity of eIF4E1b for mRNA but they could not determine significant differences in the conditions tested.

Collectively, the work presented here to address my question and the ones raised by the other reviewers expand the interest of eIF4E1b function to later stages of vertebrate embryogenesis and merits publication on EMBO Reports, as this work will serve as steppingstone for more analysis about translation regulation in the field.

We thank the reviewer for the positive feedback.

The authors could improve the final version of the manuscript addressing the following points:

- 1.- The RIP experiment that shows that eIF4E1b does not bind as many mRNAs as eIF4E1c at 3 hpf. This experiment also shows as a control a RIP experiment performed in WT embryos not expressing any FLAG protein, yet still shows RNA binding. The authors should clarify why they observe higher background binding in the control beads that with eIF4E1b. One possible reason is that the binding of eIF4E1b to the beads may block non-specific RNA binding.

Yes. As the reviewer points out, we think that eIF4E1b may have blocked unspecific binding of mRNAs to the anti-GFP beads. Importantly, unspecific binding is not relevant in our RIP-seq experiments, as a differentially expression analysis of mRNAs bound to eIF4E1b and eIF4E1c was performed.

2.- In their reply to my comment, the authors show that targets of eIF4E1b are readenylated at later stages of embryogenesis, yet it is **not clear if the targets are released from eIF4E1b and re-adenylated or just degraded**. The authors recently published a compendium of SLAM-seq data on early zebrafish embryogenesis that would help to determine if the bulk of eIF4E1b targets present at later stages are maternally provided or zygotically transcribed. If most of the signal comes from maternally provided genes, it will support the hypothesis that eIF4E1b releases its targets, allowing their translation.

Although we did not sequence the RIP samples obtained from 3 hpf embryos, RIP-seq experiments in 2-cell mouse embryos (Yang et al. 2023) show that eIF4E1b loses its ability to bind to maternal mRNAs during embryogenesis.

While it is difficult to imagine that eIF4E1b stores mRNAs only for later degradation in the embryo, we lack direct evidence for the fate of eIF4E1b mRNA targets. Comparison of eIF4E1b RIP targets at 1.25 hpf with SLAM-seq data (Bhat et al. 2023) shows that a large fraction of maternal mRNAs bound by eIF4E1b are degraded later during embryogenesis (see Fig 1 below) and thus classified as “maternal unstable”. Although this data seems counterintuitive, it is consistent with eIF4E1b protecting “unstable” mRNAs in the oocyte and early embryo from degradation, which occurs later in embryogenesis when eIF4E1b is degraded (Fig 1D). For example, replication-dependent histone mRNAs, which are eIF4E1b targets, are degraded upon translation. Thus, eIF4E1b release from histone mRNAs may promote their translation and further degradation in the embryo. While other eIF4E1b mRNA targets may follow a similar fate, future research is needed to understand the mechanisms that couple eIF4E1b release from mRNAs to their translation and/or degradation.

Figure 1. Classification of mRNAs enriched in eIF4E1b or eIF4E1c RIPs at 1.25 hpf according to their origin and stability, as determined by SLAM-seq (Bhat et al. 2023).

A mRNAs were classified based on their origin as maternal (M), zygotic (Z), or maternal and zygotic (MZ). According to their levels, mRNAs were classified as M stable, M unstable, MZ constant, MZ decreasing (M unstable + Z) or MZ increasing (M stable + Z). M stable mRNAs are overrepresented in the eIF4E1c RIP, whereas maternal mRNAs enriched in the eIF4E1b RIP are mostly unstable (M unstable + MZ decreasing). All transcripts identified in Bhat et al. are shown on the left (“All”).

B MZ constant mRNAs were further classified based on how fast maternal mRNAs were replaced by their zygotic counterparts. mRNAs with high kinetics (fast degradation and fast transcription) were overrepresented in the eIF4E1b RIP, whereas mRNAs with low kinetics were overrepresented in the eIF4E1c RIP.

Dr. Andrea Pauli
Research Institute of Molecular Pathology (IMP), Vienna Biocenter (VBC), Campus Vienna-Biocenter 1, 1030 Vienna, Austria
Campus-Vienna-Biocenter 1
Vienna 1030
Austria

Dear Dr. Pauli,

I am very pleased to accept your manuscript for publication in the next available issue of EMBO reports. Thank you for your contribution to our journal.
